# Geminin inhibits DNA replication licensing by sterically blocking CDT1-MCM2 interactions

Joshua Tomkins [1,2,3,6], Lucy V. Edwardes[1,2,6], Sarah V. Faull[1,2], Matthew Peach [1,2], Peter J. Gillespie[4,5], Vera Leber[1,2], Anna Schmidt [1,2], Halil Bounoua[1,2], Nicholas Sim [1,2], Rosa Camarillo[1,2], J. Julian Blow [4,5], Alexis R. Barr [1,2] ✉, Anna Barnard [3] ✉ & Christian Speck [1,2] ✉

DNA replication is tightly regulated to occur once per cell cycle, with the MCM2-7 helicase loaded onto replication origins only during G1-phase. In higher eukaryotes, geminin negatively regulates this process during S-, G2- and M-phases by binding the essential licensing factor CDT1. Although geminin's function is crucial for genomic stability, its inhibitory mechanism remains elusive. Here, we utilise a fully reconstituted human DNA replication licensing assay to dissect geminin's role. AlphaFold modelling provides structural insights into an N-terminal CDT1-binding helix of geminin, which proves essential for inhibition. Structural docking of the CDT1-geminin complex into the ORC-CDC6-CDT1-MCM2-7 (OCCM) assembly shows that geminin's long coiled-coil domain sterically clashes with the MCM2 C-terminus, rather than directly blocking CDT1 binding to ORC-CDC6-MCM2-7. Shortening the coiled-coil preserves geminin dimerisation and CDT1 binding but abolishes inhibition, confirming its mechanistic role. Surprisingly, geminin is not able to fully inhibit DNA licensing. However, CDK1/2-cyclin A can partially inhibit DNA licensing and, in conjunction with geminin, result in a complete block. These findings uncover geminin's steric inhibitory mechanism and suggest that a dual CDK-geminin axis controls human DNA replication.

Genomic stability is maintained through mechanisms that ensure complete duplication of the entire genome once per cell cycle and through correct separation of genetic material into the two daughter cells upon division[1]. Incomplete DNA replication and replicating the genome more than once both lead to severe genomic instability and can promote cancer[2]. Mutations in factors associated with the initial steps in DNA replication fork assembly result in Meier−Gorlin Syndrome, a rare genetic disorder that causes growth defects and distinctive facial features[3,4]. For these reasons, DNA replication needs to be tightly regulated.

The replicative helicase, MCM2-7, is initially loaded onto DNA at sites of potential replication initiation in the late M- and early G1-phases of the cell cycle in an inactive form prior to activation during S-phase[5]. Helicase loading is also termed pre-replicative complex (pre-RC) assembly or origin/DNA replication licensing. Pioneering work in budding yeast has revealed that this process is a multi-step reaction

[1]Institute of Clinical Sciences, Faculty of Medicine, Imperial College London, London, UK. [2]MRC Laboratory of Medical Sciences (LMS), London, UK. [3]Department of Chemistry, Molecular Sciences Research Hub, Imperial College London, London, UK. [4]Division of Molecular, Cell and Developmental Biology, School of Life Sciences, University of Dundee, Dundee, UK. [5]School of Biological Sciences, University of East Anglia, Norwich Research Park, Norwich, UK. [6]These authors contributed equally: Joshua Tomkins, Lucy V. Edwardes. ✉e-mail: a.barr@lms.mrc.ac.uk; a.barnard@imperial.ac.uk; chris.speck@imperial.ac.uk

with specific reaction intermediates[6–9]. Moreover, recent advances by us and others have allowed the establishment of a reconstituted human pre-RC assay[10–12]. In both organisms, the Origin Recognition Complex (ORC) initially binds to DNA, which is followed by binding of CDC6[13–17]. ORC-CDC6 then recruits MCM2-7 and CDT1 to form the ORC-CDC6-CDT1-MCM2-7 (OCCM) intermediate, which leads to the insertion of DNA into the MCM2-7 ring[10–12,18]. Consequently, the reaction proceeds in an ATPase-dependent manner, releasing CDC6 and CDT1 to generate an MCM2-7-ORC (MO) intermediate that recruits a second MCM2-7 hexamer[8,11,12]. In the human system, ORC6- and MO-independent helicase loading is possible, although it is less efficient[10–12]. The final product of the reaction cascade is a salt-stable MCM2-7 double hexamer that encircles double-stranded DNA[6,7,10–12,19] (Fig. 1a and Supplementary Fig. 1).

Controlling the temporal and spatial regulation of DNA replication licensing is essential to strictly enforce once-per-cell cycle replication. Human CDT1 protein levels oscillate during the cell cycle, with high levels in G1, low levels in S-phase, and increasing concentrations during G2-/M-phase[20]. Multiple mechanisms exist to control CDT1 protein levels and therefore modulate CDT1 function. Ubiquitin-mediated degradation of CDT1 blocks DNA replication licensing in S-phase, phosphorylation of CDT1 by CDK1/cyclin A blocks MCM2-7 recruitment and promotes CDT1 degradation in G2-phase, and finally, the binding of CDT1 by the inhibitory protein geminin in S- and G2-phase inhibits DNA replication licensing[21–26]. In summary, several overlapping mechanisms prevent helicase loading during the S-/G2-phase[27] (Supplementary Fig. 2a). Crucially, in G1-phase, these mechanisms are suppressed, allowing CDT1 to function in human helicase loading[28]. It has been proposed that geminin and CDT1 form a complex in S-/G2-phase, which blocks replicative helicase loading. Interestingly, both the overexpression of CDT1 and the removal of geminin lead to genome instability and cancer, indicating that it is essential for DNA replication control to keep CDT1 protein levels in check[29–32].

Geminin is a key inhibitor of eukaryotic DNA replication. It is highly conserved among eukaryotes, but does not exist in budding yeast, where CDK is the main negative regulator[33]. Geminin is known to form a dimer via its coiled-coil domain (Supplementary Fig. 2b, c), which assembles rapidly after translation[34]. In human cells, CDT1 is recruited to chromatin in G1-phase and peaks in late G1-/early S-phase before it is degraded, while geminin becomes recruited in early S-phase and peaks on chromatin in late S-/early G2-phase[35,36] (Supplementary Fig. 2a). CDT1 can bind to chromatin independently of geminin, while geminin's chromatin interaction is CDT1-dependent[37]. However, human geminin recruitment to DNA has never been investigated using a fully reconstituted DNA replication licensing assay. The recruitment mechanism of geminin to DNA has been investigated in vitro using a reconstituted Xenopus system. Initial studies showed that Xenopus CDT1 and geminin interact and that upon geminin addition, geminin binds to chromatin in a CDT1-dependent fashion[38]. It was found that the addition of geminin to Xenopus egg extracts blocked MCM2-7 recruitment and DNA replication, while another study found that the addition of a CDT1-geminin complex to a CDT1-depleted extract was competent for MCM2-7 loading and DNA synthesis[39]. Moreover, it has been suggested that a Xenopus CDT1-geminin hetero-trimer (1:2 CDT1:geminin ratio)[39] would be proficient for helicase loading, while a hetero-hexamer (2:4 CDT1:geminin ratio, a head-to-tail dimer of the hetero-trimer, Supplementary Fig. 2d)[40] would block MCM2-7 recruitment and DNA replication. A permissive Xenopus tetramer (1:3 CDT1:geminin ratio)[39] and inhibitory pentamer (1:4 CDT1:geminin ratio)[39] have also been proposed. These studies employed a highly complex Xenopus extract, and therefore, it remains a possibility that other factors may have modulated the CDT1-geminin interactions. For these reasons, it is not understood how Xenopus or human geminin regulates CDT1.

The domain organisation of human geminin and CDT1 has been established based on structures[21,41], which show that a CDT1 middle helical domain (CDT1[MHD]) interacts with geminin (Supplementary Fig. 2b, c). NMR has revealed that the CDT1 C-terminal helical domain (CDT1[CHD]) binds MCM6[42]. Since CDT1's MCM2-7 interaction domain and geminin's MCM2-7 interaction domain do not overlap (Supplementary Fig. 2b), it has been unclear how geminin inhibits MCM loading. Interestingly, the structure of the yeast OCCM revealed that the yCdt1[MHD] also interacts with yMCM2-7[43,44]. Therefore, a geminin interaction with CDT1[MHD] could block the MCM2-7-CDT1[MHD] interaction, but this has never been tested. Furthermore, a MCM2-7-CDT1[MHD] interaction could not explain how geminin blocks the CDT1[CHD] interaction with MCM2-7, which initiates complex assembly in budding yeast[45]. In budding yeast, yCdt1 and yMCM2-7 form a stable complex[46,47], but human CDT1 does not interact with purified MCM2-7 in solution[10]. It has been shown that the yCdt1[CHD] domain remodels yMcm6 to relieve an autoinhibitory domain and facilitate yORC-yCdc6-dependent recruitment of yCdt1-yMCM2-7[45]. However, how human CDT1 is recruited to the pre-RC and how the individual CDT1 domains contribute to this is not well understood. Therefore, it is unknown how geminin can block the CDT1-MCM2-7 interaction.

In summary, how geminin mechanistically inhibits DNA replication licensing remains poorly understood. Here, we have employed a fully reconstituted assay to address how geminin inhibits human pre-RC formation. We asked how far helicase loading can proceed in the presence of geminin and what is the oligomerisation state of the inhibitory CDT1-geminin complex. Based on AlphaFold analysis, we addressed which sections of geminin interact with CDT1, what the binding affinities of these interactions were, and which sections of geminin were required to inhibit DNA replication licensing. This was complemented with a domain analysis of CDT1, which revealed that the CDT1[MHD] and CDT1[CHD] are both required for CDT1 function. This explains why geminin's interaction with the CDT1[MHD] is sufficient to block pre-RC formation. Crucially, by combining our analysis with our recent cryo-EM structure of the OCCM[10], we identify the molecular basis of geminin-mediated inhibition of DNA replication licensing. Finally, we demonstrate that full inhibition of DNA replication licensing needs CDK2 or CDK1 activity in addition to geminin, revealing a dual mechanism of inhibition.

## Results

### Geminin inhibits DNA replication licensing in a reconstituted assay

During the last few years, the field has been hampered by the lack of a fully reconstituted human pre-RC formation assay to address how the reaction can be regulated. The recent reconstitution of human pre-RC assembly has begun to unravel the mechanistic details of human helicase loading[10–12]. In our assay, full-length ORC1-5, ORC6, CDC6 and MCM2-7 are used, while CDT1 is missing the unstructured N-terminus (CDT1[ΔN], residues 158–546), as it impacts the yield of the purification and is not required for helicase loading[10]. Initially, all proteins are incubated in an ATP-containing buffer in the presence or absence of geminin. The preincubation step supports nucleotide binding and CDT1[ΔN]-geminin complex formation. Then, magnetic beads with immobilised human-origin DNA are added, and the mixture is incubated (Fig. 1b). Following a low-salt (LS) or high-salt (HS) wash, the DNA-bound proteins are eluted with DNaseI. Under low-salt conditions, reaction intermediates such as ORC1-5, OCCM and MO remain bound to DNA (Fig. 1c, lane 1). The addition of 300 mM NaCl to the washes (high-salt (HS), referred to as loading) removes ORC1-6, CDC6 and CDT1[ΔN], whilst enriching for loaded MCM2-7 single hexamers and the reaction end-product—the MCM2-7 double hexamer (Fig. 1c, lane 3)[10]. We asked whether adding geminin to the assay can inhibit pre-RC formation in vitro. We did not observe geminin incorporation into the complex. Moreover, in comparison to the control reaction, we

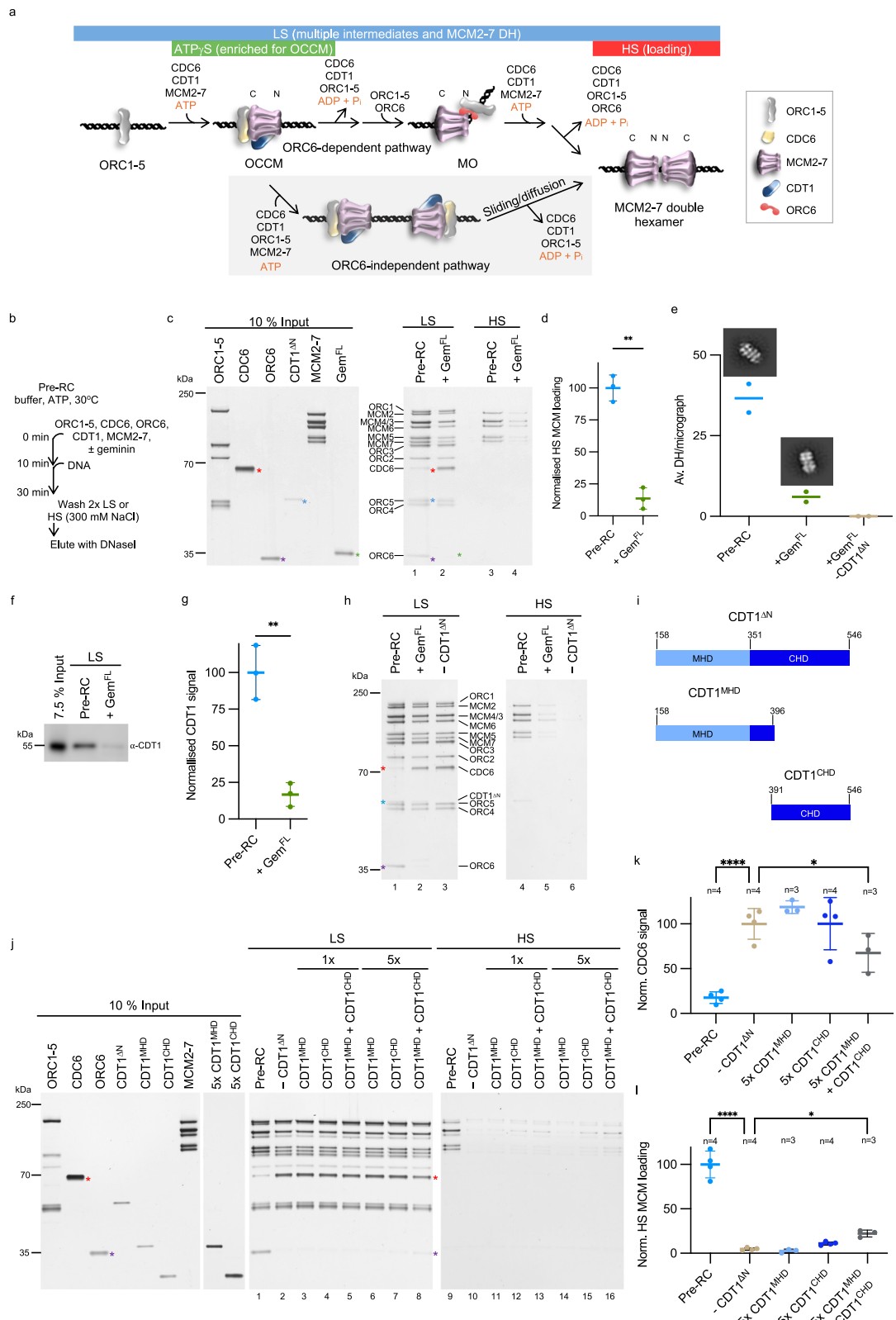

observed no recruitment of ORC6 onto DNA, although ORC1-5 were present, whilst CDC6 was stabilised (Fig. 1c, lanes 1 and 2), even when the amount added to the assay is reduced by 50% (Supplementary Fig. 3a). As CDC6 release and ORC6 recruitment are linked to ATP-hydrolysis[10], we suggest that geminin blocks DNA replication licensing prior to pre-RC ATP-hydrolysis. Consistently, geminin was able to inhibit helicase loading in the absence of ORC6 as well (Supplementary

Fig. 3b). Consequently, high-salt washes showed reduced salt-stable MCM2-7 loading in the presence of geminin (Fig. 1c, lane 4). Our quantification identified a reduction of high salt-stable MCM2-7 loading by ~85% (Fig. 1d).

To determine whether reduced MCM2-7 loading was due to decreased single or double hexamer assembly, we used negative stain electron microscopy (EM). We assembled the pre-RC reaction in

**Fig. 1 | Inhibition of DNA licensing by geminin in an in vitro assay. a** Simplified cartoon of human DNA licensing. ATP binding and hydrolysis highlighted in orange. The addition of ORC6 in our reactions means that the ORC6-dependent pathway is expected to be dominant. Coloured bars identify complexes associated with each pre-RC assay wash type. **b** Schematic of pre-RC assay protocol. **c** Pre-RC assay with reactions assembled ± geminin (Gem$^{FL}$). Gem$^{FL}$ (*green) stabilises CDC6 (*red) and inhibits CDT1$^{\Delta N}$ (*blue) and ORC6 (*purple) interaction within LS elutions. With HS elutions, Gem$^{FL}$ inhibits MCM2-7 loading. **d** Quantification of MCM2-7 loading in the HS-reactions of (**c**, lanes 3 and 4). Data normalised to the mean pre-RC reaction and analysed by two-tailed paired t-test (**p = 0.0042). Mean of three biological repeats ±SD. **e** In solution EM analysis of DH formation. 500–1000 micrographs were collected for each condition, and particles picked automatically. Two rounds of 2D classification were used to select for DHs (representative classes shown) and used to determine DH/micrograph. Mean of two independent reactions. **f** Anti-CDT1 western blot of LS pre-RC reactions ± Gem$^{FL}$. **g** Quantification of CDT1 signal in (**f**).

Data expressed as a percentage of the CDT1 input, normalised and compared to the control pre-RC mean and analysed by two-tailed paired t-test, **p = 0.0052. Mean ± SD of three biological repeats. **h** Exclusion of CDT1$^{\Delta N}$ from pre-RC reactions inhibits MCM2-7 loading (*red = CDC6, *blue = CDT1$^{\Delta N}$ and *purple = ORC6). Representative of three biological repeats. **i** Domain organisation of our standard CDT1$^{\Delta N}$ construct (residues 158–546), the CDT1 middle helical (CDT1$^{MHD}$, residues 158–396) and C-terminal helical (CDT1$^{CHD}$, residues 391–546) domains. **j** Pre-RC assay of the CDT1 MHD and CHD at 1× and 5× concentration (compared to CDT1$^{\Delta N}$ control). **k** Quantification of CDC6 signal in (**j**). Reactions compared to the −CDT1$^{\Delta N}$ control using one-way ANOVA with mixed-effects analysis and Dunnett's multiple comparisons test. ****p < 0.0001 and *p = 0.0158. Mean ± SD. **l** Quantification of HS MCM2-7 loading in (**j**). Reactions compared to the -CDT1$^{\Delta N}$ control using one-way ANOVA with mixed-effects analysis and Dunnett's multiple comparisons test. ****p < 0.0001 and *p = 0.0319. Mean ± SD of three biological repeats. Source data are provided as a Source Data file.

solution with and without geminin and used 2D classification to count how many double hexamers were present per micrograph (Supplementary Fig. 4). Our EM analysis revealed a similar drop (~83%) to SDS-PAGE quantification in MCM2-7 double-hexamer formation when geminin was added to the reaction. Furthermore, we did not observe any double hexamers when we added geminin to the pre-RC reaction in the absence of CDT1$^{\Delta N}$ (Fig. 1e and Supplementary Fig. 4g, h). Geminin, therefore, significantly reduces the loading of MCM2-7 double hexamers onto DNA, but does not completely block it.

CDT1 is released in an ATP-dependent manner during pre-RC formation[10–12]. Thus, it is challenging to detect CDT1 in pre-RC assays, compounded by the proximity of the CDT1$^{\Delta N}$ band to the ORC4/ORC5 doublet bands (Fig. 1c, lane 1, blue asterisk). To determine the effect of geminin on CDT1$^{\Delta N}$ levels, we used immunoblotting (Fig. 1f). Quantification revealed that the presence of geminin reduced CDT1$^{\Delta N}$ levels by ~80% (Fig. 1g), consistent with the reduction in MCM2-7 seen in high-salt reactions and by EM analysis (Fig. 1c–e). Recently, we have shown that in the absence of CDT1, MCM2-7 recruitment under low-salt conditions can still occur but not lead to the loading of salt-stable MCM2-7[10]. Moreover, in the absence of CDT1, ATP-hydrolysis is blocked, with ORC6 not recruited and CDC6 not released. To understand how the removal of CDT1 and the addition of geminin compare, we performed side-by-side reactions. Under low-salt conditions, we observed both a loss of ORC6 recruitment and a stabilisation of CDC6 when CDT1$^{\Delta N}$ was omitted (Fig. 1h, lane 3), and we observed nearly identical results in the presence of geminin (Fig. 1h, lane 2). High-salt washes showed that salt-stable MCM2-7 loading was inhibited in the context of geminin and when CDT1$^{\Delta N}$ was omitted (Fig. 1h, lanes 5 and 6). Since we saw with geminin an enrichment of CDC6 and absence of ORC6 (Fig. 1c, lane 2), our data suggest that geminin slows down helicase loading at the ORC-CDC6-MCM2-7 stage, just before CDT1 recruitment, OCCM formation and ATP hydrolysis-dependent MCM2-7 ring closure (Fig. 1a and Supplementary Fig. 1).

## The CDT1$^{MHD}$ and CDT1$^{CHD}$ are both required for high-salt stable MCM2-7 loading

To understand how geminin inhibits CDT1, it is essential to initially define which domain of CDT1 is actually participating in helicase loading. Recently, we and others showed that the unstructured CDT1 N-terminus is not required for DNA licensing in vitro[10–12]. Now, we asked whether the MHD or CHD of CDT1 participates in MCM2-7 loading (Fig. 1i). We expressed and purified the CDT1$^{MHD}$ and CDT1$^{CHD}$ and added them to the DNA replication licensing assay (Fig. 1j). We observed in low salt conditions that CDT1$^{MHD}$ was not recruited to the pre-RC (Fig. 1j, lane 3) and that CDT1$^{MHD}$ had no effect on MCM2-7 levels compared to the condition lacking CDT1 (Fig. 1j, lanes 2 and 3). We did not observe CDC6 release, similar to the reaction lacking CDT1 (Fig. 1j, lanes 2 and 3, Fig. 1k), suggesting that pre-RC formation is blocked before ATP hydrolysis-dependent CDC6 release. Increasing the

concentration of CDT1$^{MHD}$ had no further impact on CDT1$^{MHD}$ and MCM2-7 levels under low-salt conditions (Fig. 1j, lanes 3 and 6). Moreover, high-salt stable MCM2-7 loading was suppressed (Fig. 1j, lanes 11 and 14 and Fig. 1l). Analysis of CDT1$^{CHD}$ at 1× concentration revealed MCM2-7 association after a low-salt wash, similar to the condition without CDT1, and no stable CDT1$^{CHD}$ binding to the pre-RC (Fig. 1j, lanes 2, 4 and 7). At 5× concentration of CDT1$^{CHD}$, we did not observe CDC6 release (Fig. 1j, lane 7 and Fig. 1k), indicating an arrest before ATP-hydrolysis. The high-salt wash of the CDT1$^{CHD}$ 5× condition showed slightly more MCM2-7 loading than in the reaction lacking CDT1 (Fig. 1j, lanes 15 and 10), but the difference was not statistically significant (Fig. 1l). After a low-salt wash, a reaction containing both CDT1$^{MHD}$ and CDT1$^{CHD}$ combined resulted in a complex assembly similar to the reaction lacking CDT1 (Fig. 1j, lanes 2, 5 and 8). At 5× concentration, CDT1$^{MHD}$ combined with CDT1$^{CHD}$ had a statistically significant effect on CDC6 release (Fig. 1k), suggesting that the reaction supported limited ATP-hydrolysis and progression past the OCCM stage (Fig. 1a and Supplementary Fig. 1). Consistently, the high-salt wash reaction of the 5× CDT1$^{MHD}$ and CDT1$^{CHD}$ condition (Fig. 1j, lane 16) resulted in a small but significant increase in MCM2-7 loading compared to the minus CDT1 control. The level of loading was ~22% of the pre-RC control reaction (Fig. 1l). The data show that CDT1$^{MHD}$ and CDT1$^{CHD}$ alone have no significant activity, while combining the MHD and CHD can promote weak, high salt-stable MCM2-7 loading. We speculate that due to the slight increase in MCM2-7 levels observed with CDT1$^{CHD}$ compared to CDT1$^{MHD}$, that the C-terminus may make first contact with MCM2-7. We suggest that successful pre-RC formation requires both the CDT1 MHD and CHD domains to be linked together, consistent with a cooperative mechanism, whereby one domain binds and aids attachment of the other. This combination of two weak CDT1 interaction motifs is ideal for its function, as CDT1 only temporarily binds to MCM2-7 and has to be released upon ATP-hydrolysis to free-up the surface for other helicase activation factors in S-phase. In conclusion, the data indicate that geminin could inhibit DNA licensing by blocking either the CDT1 MHD or CHD, as both are required for its function.

## The CDT1-geminin hetero-trimer is competent for pre-RC inhibition

Previously, it was shown in the context of a *Xenopus* egg extract assay that a preformed CDT1-geminin complex is proficient for DNA replication licensing in CDT1-depleted extract[39], while others observed the opposite[48]. To assess the proficiency of a preformed complex in the reconstituted human system, which lacks other factors that could modulate CDT1 and geminin, we co-purified a CDT1$^{FL}$ and geminin complex (CG). Interestingly, the addition of CG to a CDT1-deficient pre-RC reaction resulted in similar levels of inhibition as adding geminin to the standard pre-RC reaction (Fig. 2a, lanes 5 and 6). Thus, in the reconstituted human system, a preformed CDT1$^{FL}$-geminin

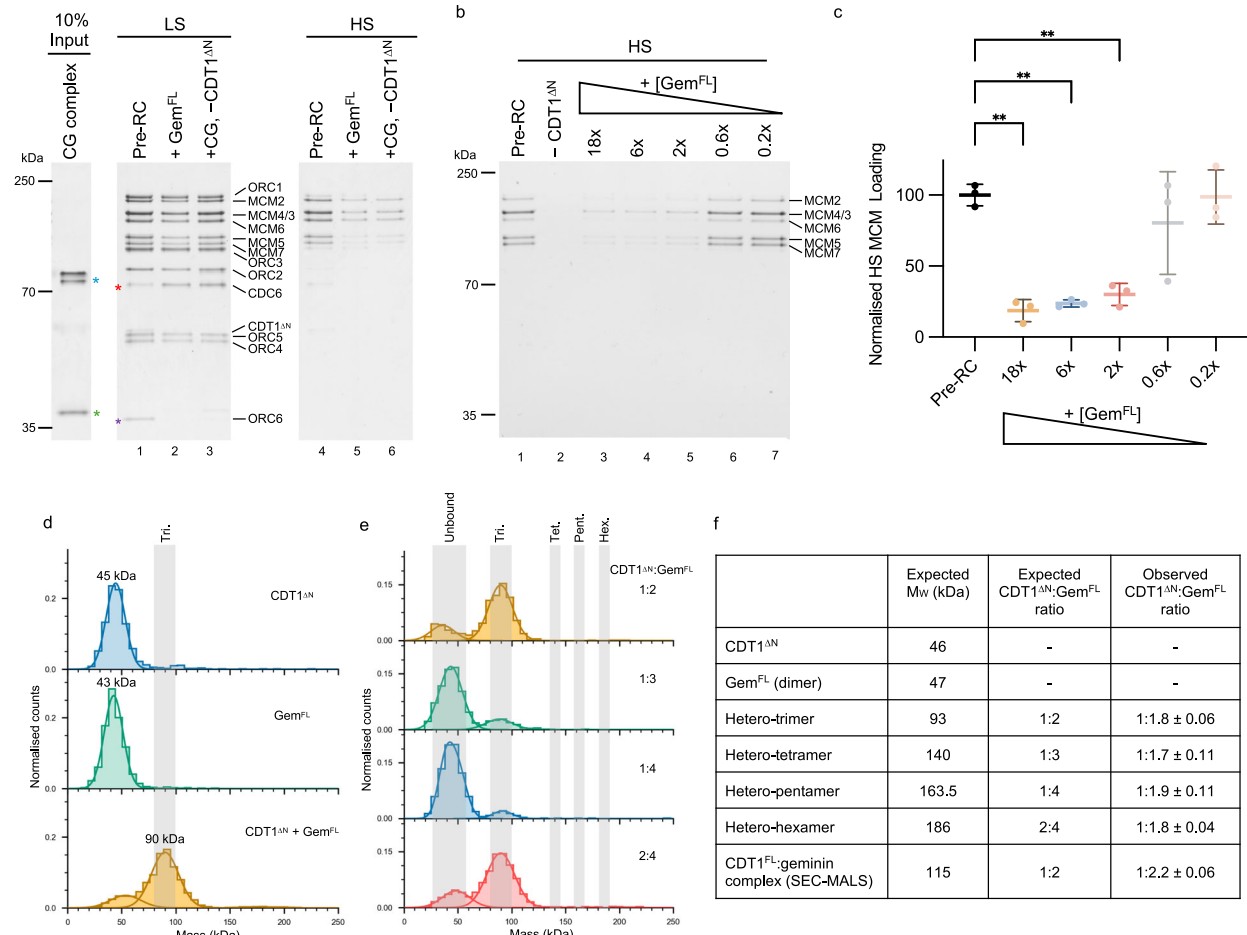

**Fig. 2 | Investigating the oligomerisation state of the CDT1-geminin complex.**
**a** Substitution of CDT1 with a co-expressed CDT1-geminin complex (+CG, −CDT1ᐞᴺ, lanes 3 and 6) inhibits pre-RC formation to a comparable extent as adding geminin (Gemᶠᴸ, lanes 2 and 5). *red = CDC6, *purple = ORC6, *green = geminin, *blue = CDT1. CDT1 runs as a doublet due to proteolytic degradation. Representative of three biological repeats. **b** Titration of geminin into the pre-RC assay, showing that at least a 2× excess (1:2 CDT1ᐞᴺ:Gemᶠᴸ = CDT1 monomer to a geminin dimer) is required to inhibit DNA licensing. **c** Quantification of MCM2-7 loading from (**b**). No significant change in inhibition is observed when adding >2× geminin. Shown as the mean ± SD of three biological repeats, normalised to the pre-RC control and analysed using one-way ANOVA with Dunnett's multiple comparisons test. **p = from bottom to top; 0.0011, 0.0017, 0.0032. **d** Mass photometry measurements of CDT1ᐞᴺ (blue), geminin (Gemᶠᴸ, green) and CDT1 + geminin (CDT1ᐞᴺ + Gemᶠᴸ, yellow) showing that CDT1ᐞᴺ and geminin form a complex of ~90 kDa, that is

consistent with the hetero-trimer (Tri., indicated by the grey shading). **e** Mass photometry exploring previously reported CDT1:geminin ratios (1:2 = hetero-trimer (yellow), 1:3 = hetero-tetramer (green), 1:4 = hetero-pentamer (blue), 2:4 = hetero-hexamer (red)). An excess of geminin results in reduced relative counts of the CDT1-geminin complex due to saturation of the movie frames with uncomplexed geminin dimers. The grey shading indicates the expected masses of each oligomeric state (abbreviated as Tri, Tet., Pent. and Hex.). **f** Expected molecular weights (Mᵂ) of CDT1ᐞᴺ, geminin and oligomeric complexes at the ratios detailed. Observed Mᵂ obtained from mass photometry measurements in parts (**d**, **e**) and from SEC-MALS of the CG complex from part (**a**). Geminin forms a dimer in solution (Mᵂ 48 kDa), and a hetero-trimer (1:2 CDT1ᐞᴺ:geminin) is observed by mass photometry and SEC-MALS under all ratios tested. Data representative of two biological repeats and shown as the mean ± SE. Source data are provided as a Source Data file.

complex blocks helicase loading, identical to when geminin is added to a standard pre-RC reaction.

It has been proposed that a CDT1-geminin heterotrimer (1:2 CDT1:geminin ratio) may form at low concentrations and support pre-RC formation, whereas high protein concentrations could result in the formation of an inhibitory CDT1-geminin hetero-hexamer (2:4 CDT1:geminin)[40] (Supplementary Fig. 2d). The existence of a permissive tetramer (1:3 CDT1:geminin)[39] and inhibitory pentamer (1:4 CDT1:geminin)[39] have also been suggested. We performed a geminin titration to test the impact of varying geminin concentrations on our pre-RC assay. We observed that sub-stoichiometric concentrations of the geminin dimer were unable to block helicase loading (Fig. 2b, lanes 6 and 7). However, low equimolar geminin dimer:CDT1ᐞᴺ monomer concentrations are competent for inhibition, while elevated geminin concentrations did not significantly increase geminin inhibition (Fig. 2b, lanes 3–5 and Fig. 2c). To address the question of the

CDT1ᐞᴺ:geminin oligomerisation state in our assay, we first measured the complex stoichiometry at the 1:2 CDT1ᐞᴺ:geminin ratio using mass photometry. At this ratio, we observed the formation of a complex with a molecular weight ~90 kDa, which is consistent with the CDT1ᐞᴺ-geminin hetero-trimer, and reduction of the peak corresponding to the molecular weights of CDT1ᐞᴺ and geminin. We did not, however, detect the formation of higher molecular weight complexes, such as the hetero-hexamer (Fig. 2d). We next increased the concentration of geminin in the assay to mimic the ratio of the proposed hetero-tetramer (1:3 CDT1ᐞᴺ:geminin)[39] and hetero-pentamer (1:4 CDT1ᐞᴺ:geminin)[39] (Fig. 2e, green and blue histograms). Increasing the ratio of geminin compared to CDT1ᐞᴺ did not drive the formation of additional higher-order CDT1ᐞᴺ-geminin complexes beyond the hetero-trimer, but instead increased the counts of uncomplexed geminin/CDT1ᐞᴺ. As increasing the ratio of CDT1ᐞᴺ:geminin past 1:2 resulted in apparent saturation of CDT1, we then doubled the concentration of the

sample where we first detected the hetero-trimer (Fig. 2d) to mimic the hetero-hexamer ratio of 2:4 CDT1$^{\Delta N}$:geminin. Under these conditions, the complex detected was again consistent with the molecular weight of the hetero-trimer ($89 \pm 1$ kDa), with no higher-order assemblies detected (Fig. 2e, red histogram). Finally, we endeavoured to detect the higher-order CDT1$^{FL}$-geminin complexes using SEC-MALS and the co-expressed CDT1$^{FL}$-geminin (CG) complex that was inhibitory in our pre-RC assay (Fig. 2a). For the SEC-MALS analysis, we used a >25× higher concentration of CDT1-geminin than in the pre-RC assay. The detected complex was, again, consistent with a hetero-trimer (Supplementary Fig. 5). Thus, we conclude that a CDT1-geminin hetero-trimer is active in pre-RC formation and that equimolar concentrations of the CDT1 monomer and geminin dimer are necessary for inhibition (Fig. 2c, f). Higher concentrations of geminin may stimulate the formation of higher-order complexes[39,40], but are not intrinsically required for the inhibition of DNA replication licensing.

## AlphaFold predictions of the hetero-trimer reveal an extended CDT1-geminin interface

Our data suggest that geminin and CDT1 form a functional hetero-trimer in solution. The structure of the mouse and human oligomers have previously been studied using crystallography employing truncated protein constructs[40]. Of the human CDT1-geminin complex, 34% of the total geminin structure (residues 91–160) and 31% of CDT1's structure (residues 167–252, 268–353) were resolved (Supplementary Fig. 2c). Since >60% of the structure was left undetermined, we used AlphaFold3[49] to predict the conformation of the full-length CDT1-geminin hetero-trimer (Supplementary Fig. 6). We also submitted a hetero-hexamer (two copies of CDT1$^{FL}$ and four copies of geminin) to the AlphaFold3 server, but none of the resultant models were consistent with the previously reported crystal structure (Supplementary Fig. 7). We therefore focused our AlphaFold analysis on the hetero-trimer, which revealed an overall geometry similar to the hetero-trimer crystal structure. AlphaFold also predicted helices that were not previously observed in both geminin and CDT1, in addition to several previously unobserved intra-molecular interactions (Fig. 3a–e and Supplementary Fig. 6d). The contact probabilities and PAE scores of these are detailed in Supplementary Fig. 6e. In particular, we observed the following regions of interest:

Region 1) In CDT1, we observed a helix (residues 113–150) interacting with the coiled-coil of geminin (residues 132–157 of chain A, and residues 143–160 of chain B), which formed a relatively large binding interface between the two proteins (Fig. 3a, b).

Region 2) The N-terminus of the geminin coiled-coil (residues 98–114) in our model is predicted to interact with residues 434–446 of the CDT1 CHD (Fig. 3a, c). This region of CDT1 has previously been observed to bind MCM2/4/6[46].

Region 3.1) This contains the coiled-coil region of geminin (residues 96–160), which forms the geminin dimer interface (Fig. 3a, d), similar to that observed in the CDT1-geminin crystal structures[40].

Region 3.2) We observed, here, a short loop in geminin (residues 76–82 of chain A) and an N-terminal adjacent short helix (residues 83–95 of chain A) that interacts with the CDT1 MHD (Fig. 3a, e). Interestingly, part of the short helix and the adjacent loop of geminin (residues 76–85) were not resolved within the human CDT1-geminin crystal structure[40].

The predicted complex provides a framework for us to explore the functional relevance of these regions.

## Region 1: a CDT1 helix is predicted to bind to the geminin coiled-coil

AlphaFold predicted an α-helix in CDT1 (residues 113–150), which interacts with the geminin coiled-coil region (Fig. 3b) and was not previously seen in the human and mouse geminin-CDT1 crystal structures[40,41]. This α-helix is reasonably well predicted

(Supplementary Fig. 6), but analysis by the ConSurf Server, which uses entries in the Protein Data Bank (PDB) to generate alignments of homologous sequences and compute conservation rates[50], highlights that it is not well conserved (Supplementary Fig. 8a). Electrostatic analysis of the interface revealed that this section of geminin contains a negatively charged patch (Supplementary Fig. 8c). The N-terminal extension (NTE) of CDT1 is unstructured (Supplementary Fig. 9), and therefore, human DNA replication licensing studies usually employ constructs lacking this domain to increase protein purification yields[10–12]. To determine whether the CDT1 N-terminus (residues 1–157) contributes to geminin-mediated inhibition, we purified CDT1$^{FL}$ (residues 1–546, including region 1) and compared it to our usual expression construct (CDT1$^{\Delta N}$, residues 158–546, missing region 1) (Fig. 3f). Deletion of the CDT1 N-terminus (residues 1–157), including the region 1 α-helix, had no impact on MCM2-7 loading or the geminin-mediated inhibition of DNA replication licensing (Fig. 3g). We therefore suggest that the helix has no functional relevance for geminin-mediated regulation of DNA replication licensing in vitro and the predicted positioning of this helix by AlphaFold can likely be attributed to the charges of the interaction surfaces (Supplementary Fig. 8c–e).

## Region 2: the CDT1$^{CHD}$ does not interact with geminin

The AlphaFold prediction suggested an interaction between geminin and the CDT1$^{CHD}$ (Fig. 3c). Considering that the CDT1$^{CHD}$ is essential for DNA replication licensing (Fig. 1j), an inhibitory CDT1$^{CHD}$-geminin interaction could be a mechanism of inhibition. To understand the role of the CDT1$^{CHD}$, we first asked whether it can interact with geminin. Using mass photometry, we observed that geminin was able to bind to the CDT1$^{MHD}$ (Fig. 3h), which is the interaction that has previously been observed in the CDT1-geminin crystal structure (Supplementary Fig. 2b, c). However, binding was not observed between geminin and the CDT1$^{CHD}$ (Fig. 3i). This suggests that geminin does not interact with the CDT1$^{CHD}$ or that the interaction is too weak to form a stable complex.

## Region 3.1: Gem$^{96-135}$ is the minimal CDT1-binding peptide

Region 3 involves the geminin coiled-coil and its N-terminal extension. Initially, we wanted to investigate the coiled-coil section of geminin (region 3.1, Fig. 4a), which was identified by structural work and AlphaFold as residues 96–160[40,49]. To address which sections of the geminin coiled-coil are relevant for CDT1 binding and inhibition of helicase loading, we synthesised a series of peptides covering the coiled-coil domain (Fig. 4b). All peptides were found to have a helical character similar to geminin when assessed by circular dichroism (CD) spectroscopy (Supplementary Fig. 10). CDT1 binding affinity was investigated by surface plasmon resonance (SPR). While full-length geminin bound to CDT1$^{\Delta N}$ with a $K_D$ of $12 \pm 3$ nM, we observed that the C-terminal peptides containing truncations in residues 135–160 retained the ability to interact with CDT1$^{\Delta N}$ with $K_D$ values in the range of ~21–38 nM. However, the C-terminal deletion of residues geminin 131–135 resulted in a complete loss of interaction with CDT1$^{\Delta N}$ (Fig. 4c, d and Supplementary Fig. 11b). Moreover, the N-terminal geminin deletion of residues 96–101 resulted in a notably weaker binding ($K_D = 2800 \pm 300$ nM). As such, we define the minimal geminin coiled-coil section that is required for CDT1 binding as residues 96–135.

The geminin coiled-coil truncations (Gem$^{96-130}$ and Gem$^{102-160}$) that failed to interact with CDT1$^{\Delta N}$ could be defective for two reasons: they could affect the CDT1-geminin interface or alternatively impact geminin dimerisation (Supplementary Fig. 11b, g). Thus, we determined the oligomerisation states of the peptides using SEC-MALS (Supplementary Fig. 12a). All peptides were found to have molecular weights consistent with dimerisation, except Gem$^{96-130}$. This peptide displayed an oligomerisation state of $1.18 \pm 0.02$, indicating that it is predominantly monomeric in solution. Gem$^{FL}$ produced a higher oligomerisation state, however, this is likely misguided by the unstructured

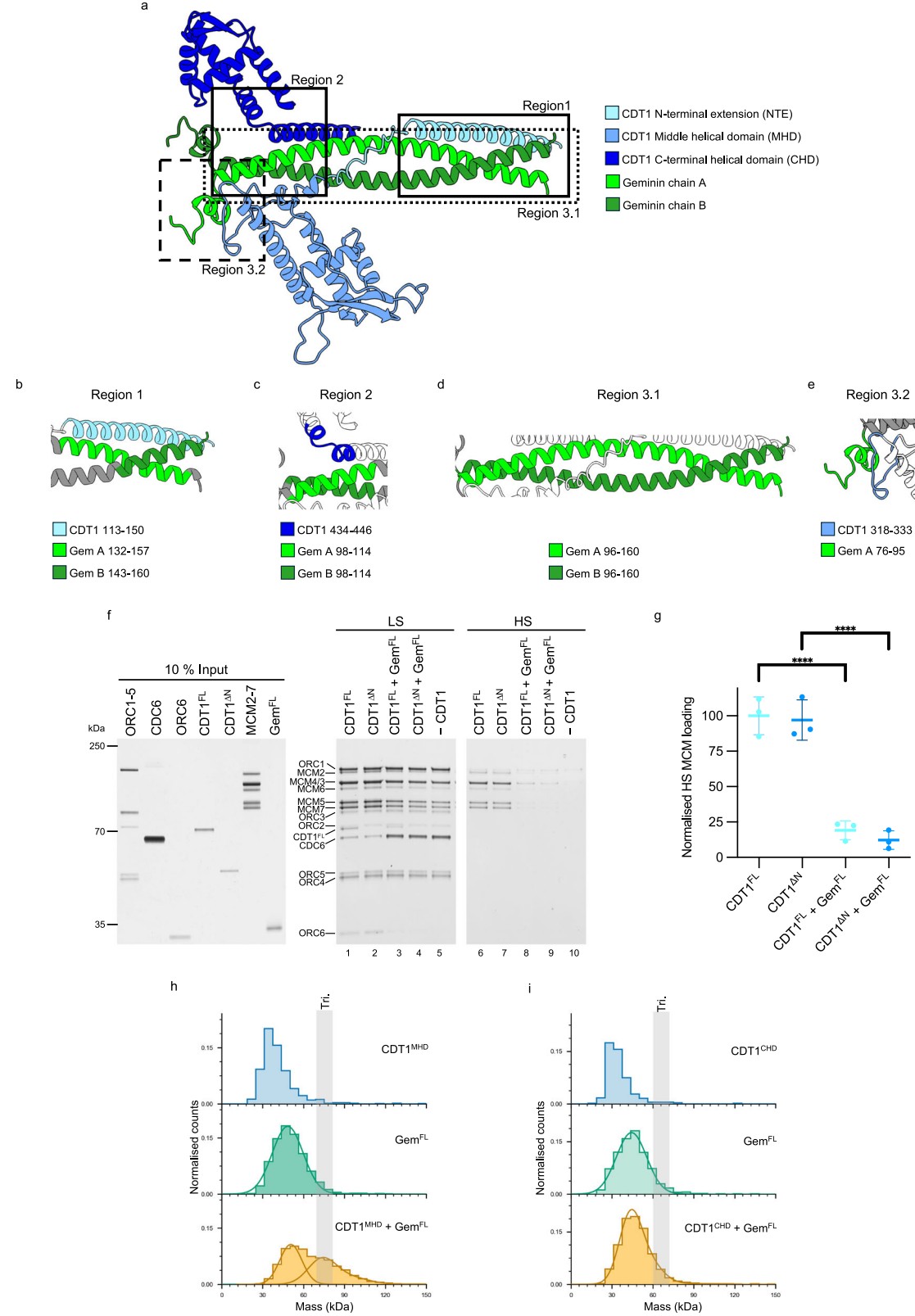

section of geminin, whereas mass photometry, which measures molecular weights independent of protein shape, identified an average oligomerisation state of 1.91 ± 0.09 (Fig. 2d). Geminin residues 131–135 are part of the leucine-zipper region and are characterised by I131 and L134 at internal positions within the coiled-coil heptad repeat (Supplementary Fig. 13a, b). As deletion of residues 131–135 led to a reduction in dimerisation (Supplementary Fig. 12a, b), we reasoned

that I131 and L134 could be important in this process. To test this hypothesis, we mutated both residues to alanines and observed ~40% reduction in MCM2-7 loading, while WT geminin resulted in a ~85% reduction in MCM2-7 loading compared to the pre-RC control reaction that did not contain geminin (Supplementary Fig. 13d, e). Moreover, as CDT1 contacts both geminin coiled-coils (Fig. 3a), we suggest that amino acids 131-135 are vital for dimerisation of the geminin coiled-coil

**Fig. 3 | AlphaFold reveals new interactions between CDT1 and geminin.**
**a** AlphaFold3 model of the CDT1-geminin hetero-trimer with CDT1 coloured blue and geminin green. Unstructured regions not shown. **b**–**e** The interaction regions identified by AlphaFold analysis. Geminin is shown in grey, with interaction regions shown in green, and CDT1 is shown in white with interaction regions in blue. Details of coloured residues are shown below the cartoons. Figures made using Chimera X. **b** Region 1: N-terminal CDT1 helix. **c** Region 2: CDT1 C-terminus with the N-terminal end of the geminin coiled-coil. **d** Region 3.1: the geminin coiled-coil **e** Region 3.2: geminin N-terminal helix with the CDT1 middle helical domain. **f** Pre-RC assay comparing full-length CDT1 (CDT1$^{FL}$, residues 1–546), which contains the region 1 helix of interest shown in (**b**), and our standard CDT1 expression construct (CDT1$^{\Delta N}$, residues 158–546) in which the helix of interest, along with the unstructured N-terminal extension, have been truncated to improve protein stability.
**g** Quantification of MCM2-7 loading under high-salt (HS) conditions shown in part

(**f**), showing no significant difference in MCM2-7 loading or inhibition of loading by geminin between the two CDT1 constructs. Data shown as the mean ± SD of three independent experiments, normalised to the mean of the CDT1$^{FL}$ reaction and analysed by one-way ANOVA with Tukey's multiple comparisons test, ****$p < 0.0001$. **h** Mass photometry of the middle helical domain (MHD) of CDT1 with geminin. Geminin results in a peak at ~48 kDa (green) and when CDT1$^{MHD}$ is added to geminin, a secondary peak emerges at ~78 kDa, which is consistent with the predicted mass of a CDT1$^{MHD}$-geminin hetero-trimer (tri., expected $M_W$ 77 kDa, yellow). The MHD ($M_W$ 29 kDa) is below the limit of detection of ~30 kDa, therefore a gaussian fit has not been applied (blue). **i** Mass photometry of the C-terminal helical domain (CHD) of CDT1 with geminin. No secondary peak (predicted $M_W$ 67 kDa) is detected suggesting that the CDT1$^{CHD}$ cannot bind to geminin. CDT1$^{CHD}$ ($M_W$ 19 kDa) is below the limit of detection of ~30 kDa, therefore a Gaussian fit has not been applied (blue). Source data are provided as a Source Data file.

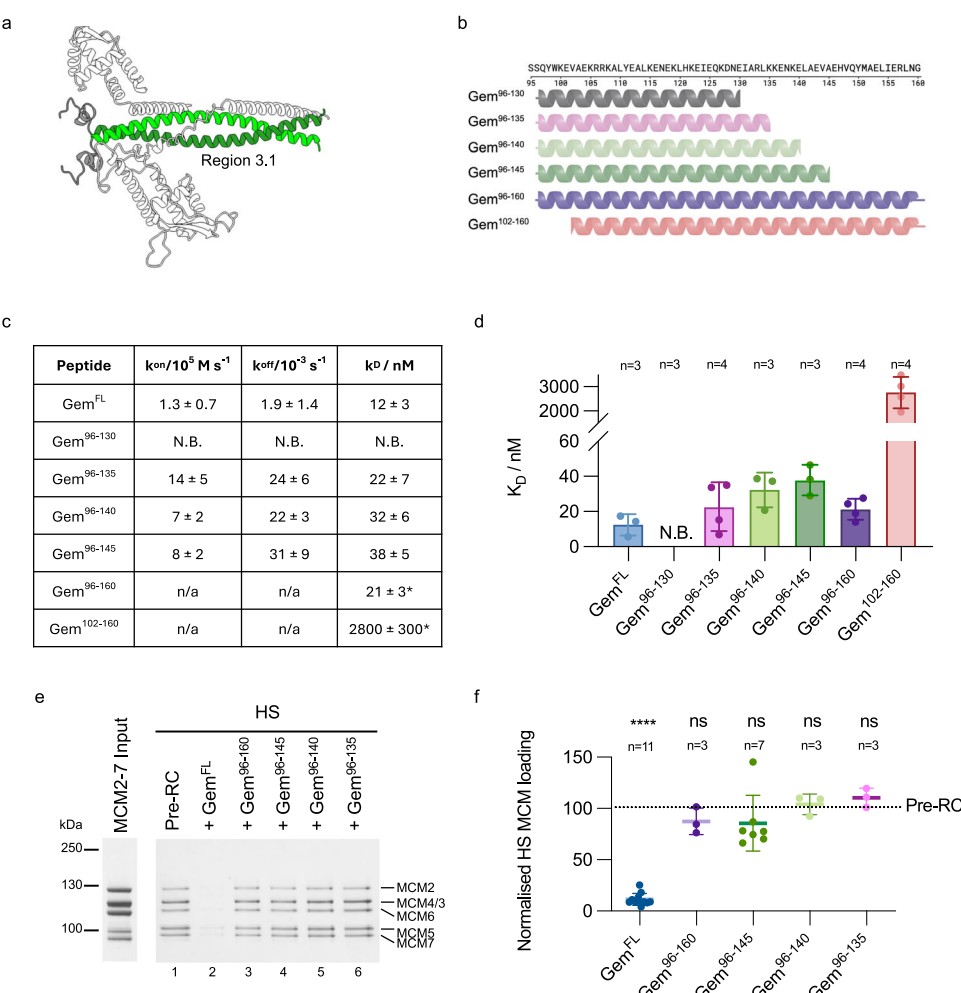

**Fig. 4 | Investigating region 3.1 interactions using peptide mimetics of the geminin coiled-coil. a** AlphaFold model of region 3.1—the geminin coiled-coil (residues of interest highlighted in green). Figure made using Chimera X. **b** Cartoon of coiled-coil peptides with the amino acid sequence above. Created in BioRender. Faull, S. (2025) https://BioRender.com/kxwwwdz. **c** Detailed kinetic and dissociation constants from SPR-determined binding data between immobilised CDT1$^{\Delta N}$ and coiled-coil mimetic peptides. *Indicates $K_D$ was determined via affinity modelling instead of kinetic fit. Number of replicates as per part (**d**), data shown as

mean ± SE. **d** Summary of SPR-determined binding data between immobilised CDT1$^{\Delta N}$ and coiled-coil mimetic peptides. Values displayed as mean ± SE. N.B. = no binding. **e** Pre-RC assay of high-salt (HS) washed reactions comparing the inhibitory activity of geminin coiled-coil mimetic peptides to full-length geminin (Gem$^{FL}$).
**f** Quantification of MCM2-7 bands from in part (**e**) compared to the control pre-RC reaction using one-way ANOVA with mixed-effects analysis and Dunnett's multiple comparisons test. ****$p < 0.0001$, ns = not significant. Data shown as the mean ± SD. Source data are provided as a Source Data file.

and, in this way, support geminin-CDT1 interactions (Supplementary Fig. 13).

Next, we used our pre-RC assay to investigate the ability of the coiled-coil peptides that robustly interact with CDT1$^{\Delta N}$ to inhibit DNA

replication licensing. Peptides Gem$^{96-135}$, Gem$^{96-140}$, Gem$^{96-145}$ and Gem$^{96-160}$ were added to a pre-RC reaction and then subjected to a high-salt wash. Full-length geminin led to a strong inhibition in helicase loading (Fig. 4e). Surprisingly, despite having CDT1$^{\Delta N}$-binding affinities

comparable to that of full-length geminin, these peptides were all unable to inhibit MCM2-7 loading when added to the pre-RC reaction, even though the peptides were used in 50x molar excess to CDT1$^{\Delta N}$ (Fig. 4e). Quantification of high-salt stable MCM2-7 relative to the control reaction (Fig. 4f) identified that Gem$^{FL}$ reduced MCM2-7 loading significantly, supporting just 11.4 ± 1.8% pre-RC formation compared to the reaction lacking geminin. On the other hand, Gem$^{96-135}$, Gem$^{96-140}$, Gem$^{96-145}$ and Gem$^{96-160}$ did not significantly inhibit MCM2-7 loading. Since Gem$^{96-160}$ contains the full geminin coiled-coil, we concluded that the coiled-coil alone is insufficient to inhibit MCM2-7 loading, which is consistent with findings with murine geminin[41].

### Region 3.2: Geminin N-terminal residues 76−90 form a CDT1-binding motif that is essential for the inhibition of DNA replication licensing

Our experiments revealed that the geminin coiled-coil is insufficient to inhibit DNA replication licensing. However, the AlphaFold prediction of CDT1$^{FL}$-geminin identified a short N-terminal helix and loop (Fig. 5a) attached to the geminin coiled-coil, while only a small section of the loop was observed to interact with CDT1 in the crystal structure (PDB ID 2WVR, Supplementary Fig. 2c)[40]. To address the role of the geminin N-terminus in DNA replication licensing, we initially performed a computational alanine scan on the AlphaFold prediction using the BudeAlaScan (BAlaS) web server[51] to analyse how mutation of individual residues to alanine could impact the interactions between CDT1$^{FL}$ and geminin. The scan highlighted hot spot residues within our AlphaFold regions of interest (Supplementary Fig. 14). In particular, ΔΔG values were highest for regions 2 and 3.2 at the N-terminus of the geminin coiled-coil, indicating a strong interaction between CDT1$^{FL}$ and geminin in this part of the complex.

After further in silico validation of the CDT1$^{FL}$-geminin heterotrimer, we designed peptides where we N-terminally truncated full-length geminin to explore the disparity between CDT1$^{\Delta N}$ binding and inhibition of MCM2-7 loading seen with coiled-coiled series of peptides in Fig. 4. We created a peptide series of Gem$^{76-145}$, Gem$^{82-145}$, Gem$^{90-145}$ and Gem$^{96-145}$ (Fig. 5b) and their oligomerisation states were determined to be predominantly dimeric by SEC-MALS (Supplementary Fig. 15a). SPR experiments showed that N-terminal extension of the geminin coiled-coil improved CDT1$^{\Delta N}$-binding (Fig. 5c, d). The strongest binding was observed with Gem$^{76-145}$ and Gem$^{82-145}$, which bound CDT1$^{\Delta N}$ with $K_D$ = 5.7 ± 0.7 nM and 5.8 ± 0.7 nM, respectively. Notably, this binding was stronger than that observed using Gem$^{FL}$ ($K_D$ = 12 ± 3 nM). Gem$^{96-145}$ bound CDT1$^{\Delta N}$ with a $K_D$ of 38 ± 5 nM and interestingly, Gem$^{90-145}$ displayed weaker CDT1$^{\Delta N}$ binding ($K_D$ = 93 ± 16 nM) despite the longer N-terminal extension compared to Gem$^{96-145}$, suggesting a potential structural defect that weakens the interaction (Supplementary Fig. 16). The $k_{off}$ rates of Gem$^{82-145}$ and Gem$^{76-145}$ decreased by around 10-fold when compared to the shorter Gem$^{90-145}$ and Gem$^{96-145}$ peptides, whilst the $k_{on}$ values were similar. The inclusion of the geminin residues identified in region 3.2, therefore, stabilises the complex. Residues 82−90 of geminin are predicted to form a helix and a short loop, which interact with CDT1. Part of the helix and the entire loop were not observed in the CDT1-geminin crystal structure, but we hypothesised that they form a key interaction interface to enable geminin to inhibit DNA replication licensing. We therefore tested the ability of our N-terminally extended peptides to inhibit DNA licensing in the pre-RC assay. Both Gem$^{76-145}$ and Gem$^{82-145}$, which contain the predicted N-terminal helix and loop, inhibited the loading of salt-stable MCM2-7 to a similar extent (25 ± 3% and 30 ± 7%, Fig. 5e, lanes 3 and 4 and Fig. 5f), approaching levels seen with full-length geminin. In contrast, Gem$^{90-145}$ and Gem$^{96-145}$ were unable to inhibit MCM2-7 binding to DNA to any significant degree (Fig. 5e, lanes 5 and 6 and Fig. 5f). In summary, our data show that geminin residues 82−90 form an interaction motif with CDT1, which is important for the inhibition of DNA replication licensing.

### Inhibition of DNA synthesis by geminin mimetic peptides in *Xenopus* egg extract

To investigate the effect of our peptides on DNA synthesis, we used a *Xenopus* egg extract assay. Assays were performed with Gem$^{76-145}$ and Gem$^{82-145}$ (inhibitors of DNA replication licensing in the pre-RC assay), Gem$^{96-145}$ (able to bind CDT1 but not inhibit DNA replication licensing in the pre-RC assay) and Gem$^{96-130}$ (unable dimerise or bind CDT1). We also included a recombinantly-expressed version of Gem$^{76-145}$, to address if the cellular quality control results in improved folding (Supplementary Fig. 15b, c). Gem$^{96-130}$ and Gem$^{96-145}$ were unable to inhibit DNA synthesis, in agreement with our pre-RC assay (Fig. 5g), whilst Gem$^{76-145}$ and Gem$^{82-145}$ inhibited DNA synthesis in a concentration-dependent manner. The addition of bacterially-expressed recGem$^{76-145}$ increased the inhibition slightly, which we attribute to the cellular machinery improving the peptide folding compared to chemical synthesis. The sensitivity of the *Xenopus* licensing assay may also contribute to the difference as Gem$^{76-145}$ and recGem$^{76-145}$ exhibited comparable levels of inhibition in the pre-RC assay (Fig. 5h). Gem$^{76-145}$ was predicted by AlphaFold to contain a longer helix and short loop, which interacts with CDT1, while Gem$^{82-145}$ contained only the longer helix. Thus, the data show that the AlphaFold-predicted interactions in region 3.2 have a functional role in pre-RC formation.

### Interaction of CDT1 with the region 3.2 geminin helix/loop motif is essential for geminin function

Since the geminin loop region encompassing residues 82−95 turned out to be essential for geminin's function, we wanted to ask whether the corresponding interaction region in CDT1 is also necessary for geminin-mediated inhibition. The BUDE alanine scan results (Supplementary Fig. 14c, d) were used to identify critical amino acids at the CDT1-geminin interface, with a focus on the CDT1 region that interacts with the geminin N-terminal helix and loop (Fig. 5i). We mutated the four residues with the highest ΔΔG score within the CDT1 loop to alanines (residues 327−330, CDT1$^{4A}$) and also generated a single point mutant of the residue with the greatest change in free energy (CDT1$^{R330A}$). Both mutants were generated on a CDT1$^{\Delta N}$ background and were able to form heterotrimeric complexes with geminin when assessed using mass photometry (Fig. 5j, k). Mutation of residues 327−330 to alanines resulted in a CDT1 construct that can load MCM2-7 but was resistant to geminin inhibition (Fig. 5l, m). R330 had the highest ΔΔG score in the BUDE alanine scan, and its mutation to alanine had the same impact as CDT1$^{4A}$, highlighting the importance of R330 for the CDT1-mediated inhibition of geminin (Fig. 5l, m). Thus, the CDT1 amino acids that interact with geminin in region 3.2 are essential for geminin-mediated inhibition, while the mutations in CDT1 do not impact pre-RC formation on their own.

### Geminin inhibits the formation of a functional OCCM intermediate

Pre-RC reactions carried out in the presence of ATP and geminin resulted in CDC6 stabilisation and reduced ORC6 recruitment (Fig. 1c), indicating that they stop the multi-step helicase loading reaction prior to ATP hydrolysis. To directly ask whether geminin blocks OCCM formation, we assembled pre-RC reactions in the presence of ATPγS, which arrests helicase loading at the recruitment (OCCM) stage[10]. In the absence of geminin, we observed the assembly of an OCCM complex, where CDC6 and CDT1$^{\Delta N}$ are present (Fig. 6a, lane 1). The overall complex appeared very similar in the presence of geminin, but CDT1 was absent, and geminin was not associated with DNA (Fig. 6a, lane 2). To visualise CDT1, we employed a fluorescently labelled version of CDT1$^{\Delta N}$ (Cy3-CDT1$^{\Delta N}$) (Fig. 6b), which highlighted the absence of CDT1. Interestingly, MCM2-7 recruitment is not affected, and a ORC1-5-CDC6-MCM2-7 complex is formed instead. Thus, our data suggests that geminin sequesters CDT1 to prevent the formation of a

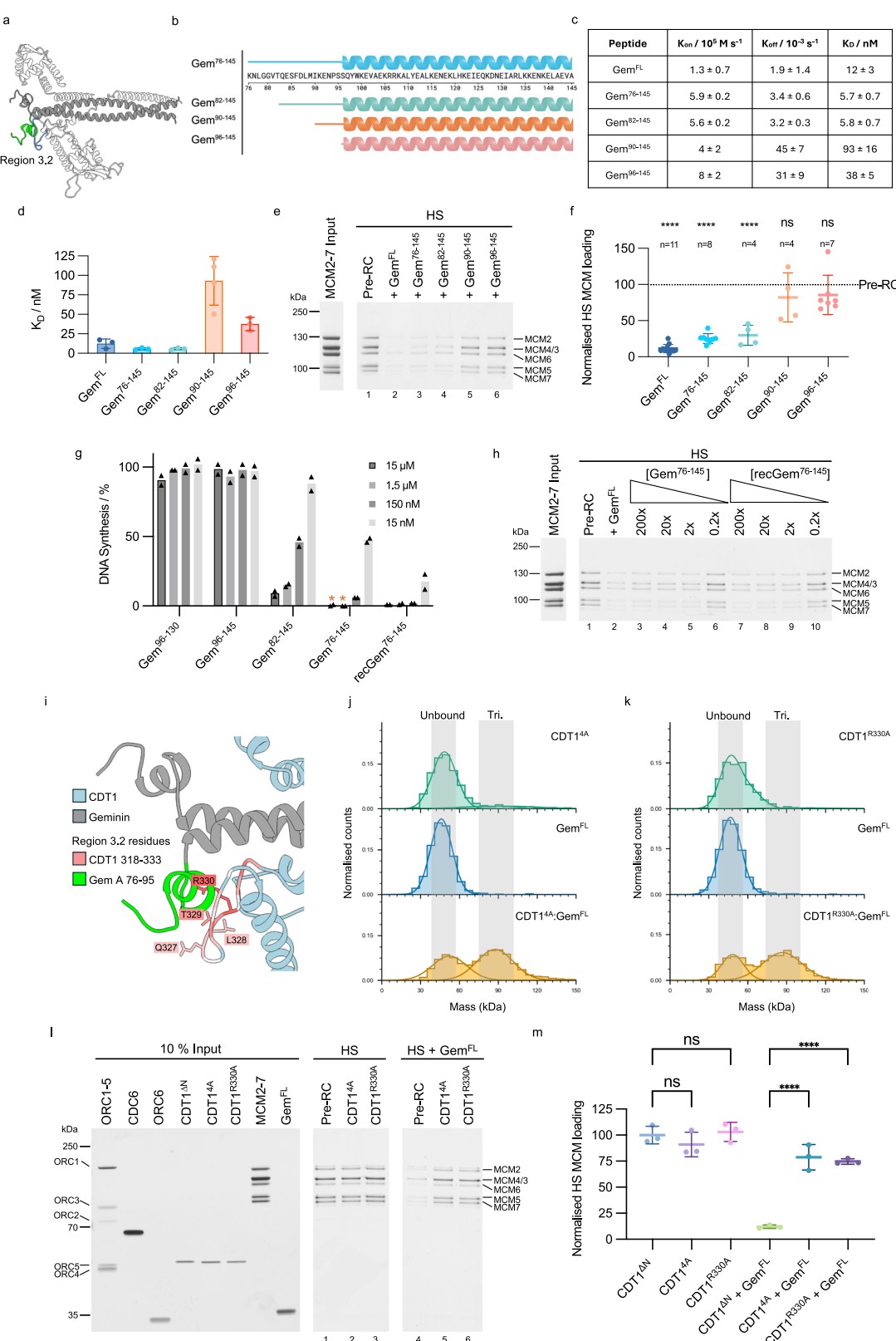

functional OCCM, which is incapable of CDT1 recruitment and CDC6 release, and therefore unable to progress to a functional MO intermediate. Next, we wondered what would happen if we challenged a preformed OCCM complex with geminin – asking the question of whether geminin can remove CDT1$^{\Delta N}$ that has already been integrated into the OCCM complex. Thus, we formed an OCCM complex in the presence of ATPγS, and then added geminin for 1, 2 or 4 min (Fig. 6c). The CDT1$^{\Delta N}$ analysis quantification highlighted that 60% of CDT1$^{\Delta N}$ was removed within 4 min, compared to a reaction lacking geminin (Fig. 6d, e). Thus, geminin can disassemble a preformed OCCM complex, and, this way, stop helicase loading even after the process has already been initiated.

**Fig. 5 | Region 3.2–Investigating the geminin N-terminus using mimetic peptides. a** AlphaFold model of interaction region 3.2. Geminin residues (76–95) shown in green and CDT1 interacting loop (318–333) in blue (Chimera X). **b** N-terminally elongated peptides. Created in BioRender. Faull, S. (2025) https://BioRender.com/whoj587. **c** Kinetic and dissociation constants from SPR binding experiments. n = 3, except for Gem$^{90-145}$ n = 4. Mean ± SE. **d** Plot of SPR-determined $K_D$ values between immobilised CDT1$^{\Delta N}$ and geminin-mimetic peptides. Mean ± SE, n as per (**c**). **e** High-salt (HS) pre-RC assays of geminin and peptide derivatives. **f** Quantification of MCM2-7 band intensity from (**e**). Mean ± SD compared to control pre-RC using one-way ANOVA with mixed-effects analysis and Tukey's multiple comparisons test (TMCT). ****p < 0.0001. ns = not significant. **g** *Xenopus* egg extract assay measuring DNA replication licensing by monitoring incorporation of [α–³²P]-dATP into DNA relative to an untreated sample. To account for baseline variation, DNA synthesis of a mock-licensed negative control sample has been subtracted. Mean of two independent experiments. * = no DNA synthesis observed. **h** Pre-RC assay comparing the ability of chemically-synthesised peptide Gem$^{76-145}$ and recombinantly expressed recGem$^{76-145}$ to inhibit DNA licencing. Concentrations refer to the ratio of monomeric geminin to CDT1, 2x represents the CDT1-geminin hetero-trimer. n = 2 biological repeats. **i** Interaction between geminin (grey with residues 76–95 in green) and CDT1 (blue with residues 318–333 coloured by ΔΔG values from the BudeAlaScan web server). Side chains with the greatest ΔΔG are shown. **j** Mass photometry of CDT1$^{\Delta N}$ in which residues 327–330 have been mutated to alanines (CDT1$^{4A}$, green). Addition of CDT1$^{4A}$ to geminin (Gem$^{FL}$, control in blue) results in a complex peak consistent with a heterotrimer (yellow. Trimer = Tri., grey shading). Representative of n = 3. **k** Mass photometry of CDT1$^{R330A}$ (green). Addition of CDT1$^{R330A}$ to geminin (Gem$^{FL}$, control in blue) results in a complex peak consistent with a heterotrimer (yellow). Representative of n = 3. **l** Pre-RC assay of CDT1 region 3.2 mutants. CDT1$^{4A}$ reduces geminin-mediated pre-RC inhibition, but mutating residue 330 (CDT1$^{R330A}$) alone is sufficient to disrupt the inhibition by geminin. **m** Quantification of HS MCM2-7 loading in (**l**). Mean ± SD of three biological repeats, compared to control pre-RC using one-way ANOVA with TMCT. ****p < 0.0001. Source data are provided as a Source Data file.

## A structural model that explains how geminin inhibits CDT1 in pre-RC formation

Our work resulted in an improved understanding of the CDT1-geminin interaction motif, while recent structural work on the human OCCM revealed the CDT1-MCM2-7 interface. Comparing the two interfaces (Supplementary Fig. 17a, b), it is clear that both motifs are not overlapping and therefore geminin binding to CDT1 should not block its interaction with MCM2-7. In turn, this means that geminin must impact helicase loading in another way. We wondered whether geminin binding would alter the structure of CDT1. Comparing the structure of CDT1 in complex with geminin or MCM2-7 did not identify any changes (Supplementary Fig. 18a, b). As the CDT1-geminin interaction stopped pre-RC formation at an early stage, prior to ATP-hydrolysis-dependent CDC6 release, we wondered whether geminin could impact OCCM formation by indirectly impacting the CDT1-MCM2-7 interaction.

Recently, the structure of the human OCCM complex was revealed[10]. This structure was obtained in the presence of ATPγS, which blocks ATP-hydrolysis. As such, the structure captured CDT1 in its active conformation, just prior to ATP hydrolysis. Here, it was observed that the CDT1$^{MHD}$ interacts with MCM2, while the CDT1$^{CHD}$ interacts with MCM4 and MCM6. Consequently, we structurally aligned the CDT1-geminin complex obtained by AlphaFold with CDT1 in the OCCM structure. Excitingly, this alignment revealed a steric clash between the geminin coiled-coil domain and MCM2 (Fig. 6f). In particular, amino acids 133–160 of geminin Chain A and 127–160 of geminin Chain B, which belong to the coiled-coil domain, were clashing with MCM2. These residues are poorly conserved at the amino acid identity level, but similarity between residues is high (Supplementary Fig. 17c). This docking experiment strongly suggests that geminin inhibits DNA replication licensing by preventing CDT1 being incorporated into the OCCM by blocking the interaction between CDT1 and the C-terminal motor domain of MCM2 (Fig. 6f, g).

## N-terminal helix/loop of geminin blocks CDT1-Geminin from binding MCM2-7

Next, we wanted to test the steric clash hypothesis. We hypothesised that geminin may position the coiled-coil domain at an angle that results in a clash with the motor domain of MCM2 (Fig. 6f). The CDT1$^{FL}$-geminin AlphaFold prediction and biochemical data argue that the geminin coiled-coil represents the primary contact point with CDT1 and that a secondary CDT1 interaction, encompassing the short geminin loop (residues 76–90), may position the geminin coiled-coil at the correct angle to generate the geminin-MCM2 clash. Consequently, we inserted a short glycine linker between the two interaction interfaces, termed Gem$^{LINK}$ (Supplementary Fig. 19a), to confer flexibility to this region and to test this model. Initially, we tested the interaction between CDT1$^{\Delta N}$ and Gem$^{LINK}$ using SPR. The $K_D$ of the CDT1$^{\Delta N}$-Gem$^{LINK}$ complex was 38 ± 4 nM (Supplementary Fig. 19b and c), while the CDT1$^{\Delta N}$-geminin complex has a $K_D$ of 12 ± 3 nM (Fig. 5c). Thus, the linker modestly impacted the interaction with CDT1.

Consequently, we used the proteins in the pre-RC assay to ask whether the insertion of a flexible linker between the two interaction sites would impact pre-RC formation. With Gem$^{LINK}$ we observed reduced recruitment of ORC6 and a mild stabilisation of CDC6 (Supplementary Fig. 19d, lane 3), which indicates that complex formation progressed past the ATP-hydrolysis step but at a reduced extent compared to the control reaction. Consistently, Gem$^{LINK}$ was less efficient in inhibiting DNA replication licensing than geminin (Supplementary Fig. 19d, lane 6). In summary, our data hint that coordination between the two CDT1-geminin interaction surfaces is needed for inhibitory activity.

## The geminin coiled-coil generates a steric clash with MCM2

To test the model of a steric clash between the geminin coiled-coil and MCM2-7 further, we reasoned that a shortening of the coiled-coil should not impact the CDT1-geminin interaction but should fail to generate the steric clash (Fig. 7a). Thus, we generated geminin peptides with reduced geminin coiled-coil length (Gem$^{76-145}$, Gem$^{76-140}$ and Gem$^{76-135}$) (Fig. 7b). Initially, we asked whether the reduced length of the coiled-coil would impact geminin dimerisation, but observed a dimeric oligomerisation state for all peptides (Supplementary Fig. 20). Next, we assessed whether the peptides retained their ability to bind to CDT1$^{\Delta N}$ using SPR. We found all peptides to have similar $K_D$ values (Fig. 7c and d), indicating that they retain their ability to interact with CDT1. Interestingly, we observed that truncation of the coiled-coil was associated with improved high-salt stable MCM2-7 loading (Fig. 7e and f). Our docking analysis revealed that amino acids 127–160 (chain A)/133–160 (chain B) of geminin clashed with MCM2. Shortening the coiled-coil to near the predicted geminin-MCM2 boundary (Gem$^{76-135}$) resulted in the recovery of 75% of pre-RC loading activity, while slightly longer geminin peptides (Gem$^{76-140}$ and Gem$^{76-145}$) resulted in incrementally less recovery (Fig. 7f). As such, our data show that geminin inhibits pre-RC formation by generating a steric clash with MCM2. To test whether the clash is sequence-specific, we inserted a short 2xGGS linker after residue 135 (Gem$^{2xGGS}$) to preserve the non-canonical coiled-coil, but alter its relative location and rotation (Supplementary Fig. 19e, f). This mutant was still functional in inhibition (Supplementary Fig. 19g, h), supporting the idea that a specific sequence is not required.

## Geminin and CDK co-operatively regulate origin licensing

The addition of geminin greatly reduces salt-stable MCM2-7 loading in our assay, however we still observe -15% MCM2-7 loading (Fig. 1c–e). This activity persists, even with excess amounts of geminin (Fig. 2b). We note that S-phase and M-phase CDK have also been postulated to regulate human DNA licensing via CDK docking sites in ORC1 and

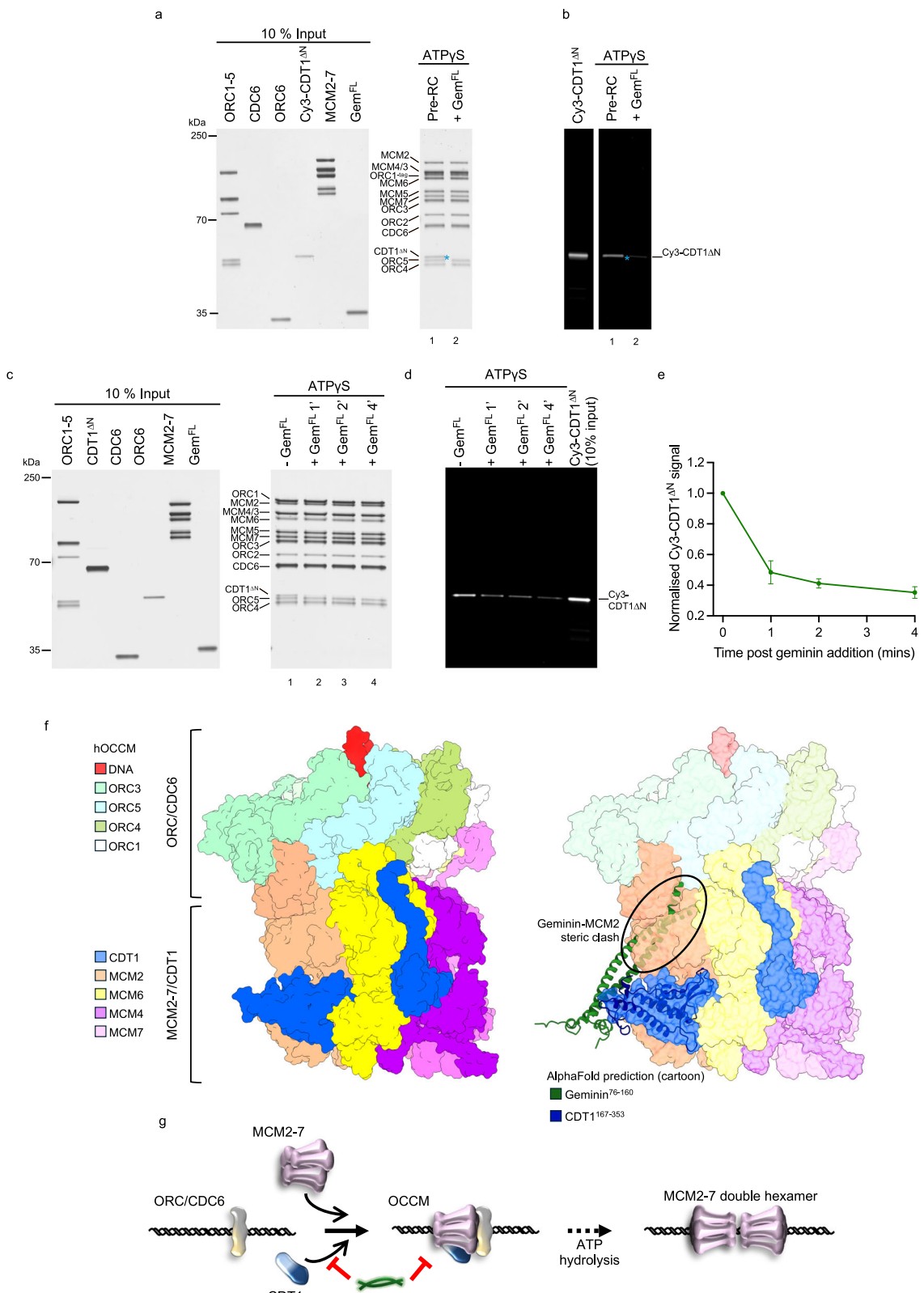

CDT1[44,52]. Moreover, in budding yeast, it is well established that S-phase CDK phosphorylation of yORC and yCdc6 inhibits DNA licensing[53]. However, whether human S-phase CDK (CDK2/cyclin A) or M-phase CDK (CDK1/cyclin A2) can inhibit DNA licensing has not been investigated using a reconstituted system.

To address which DNA licensing proteins become phosphorylated by CDK in our reconstituted system, we individually phosphorylated ORC1-5, ORC6, CDC6, CDT1[ΔN] and MCM2-7 using CDK1/cyclin A2 or CDK2/cyclin A (Supplementary Fig. 21). Using a phospho-SP antibody, we detected CDK1/cyclin A2-dependent phosphorylation of ORC1, CDC6 and CDT1[ΔN], while ORC6 was not phosphorylated. MCM2-7 showed basal phosphorylation that increased upon CDK1 treatment. Importantly, in the presence of CDK inhibitor p27, no phosphorylation was detected, confirming

**Fig. 6 | Inhibition of OCCM formation by geminin. a** Silver stained SDS-PAGE showing that the recruitment of CDT1 is inhibited by the addition of geminin to an ATPγS-containing pre-RC reaction. Representative of three biological repeats. **b** Corresponding Cy3 fluorescence emission of part (**a**) with fluorescently-labelled CDT1 (Cy3-CDT1$^{\Delta N}$, annotated with *). **c** Two-step pre-RC reaction. The OCCM was assembled in the presence of ATPγS, after 18 min the control reaction (-Gem$^{FL}$) was washed and eluted. The other reactions were spiked with geminin for either 1, 2 or 4 min before washing. **d** Corresponding Cy3 fluorescence emission of part (**c**). **e** Quantification of Cy3-CDT1 fluorescence from part (**d**) showing CDT1 dissociation

in the presence of geminin. Data shown as mean ± SD with a line of best fit of five biological repeats. **f** Alignment of CDT1-geminin AlphaFold model (shown in cartoon form) with the cryo-EM structure of the human OCCM (PDB ID 8RWV, shown in surface view) shows a steric clash between the C-terminus of the geminin coiled-coil (green) with MCM2 (peach). Figure made using Chimera X. **g** Proposed mechanism of how geminin inhibits DNA licensing by sequestering CDT1 away from the OCCM and preventing CDT1 from binding via steric hindrance. Source data are provided as a Source Data file.

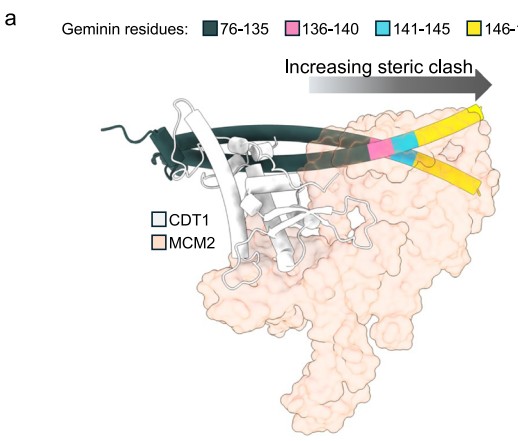

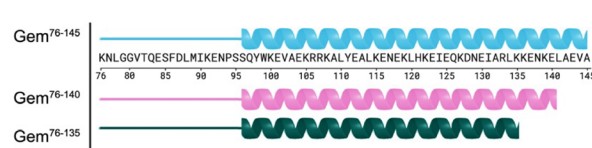

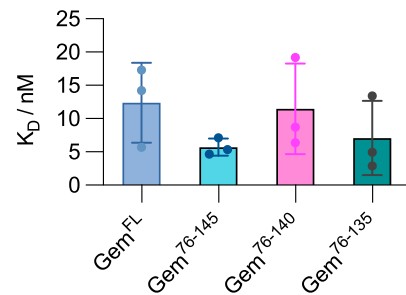

| Peptide | $K_{on}$ / $10^5$ M s$^{-1}$ | $K_{off}$ / $10^{-3}$ s$^{-1}$ | $K_D$ / nM |
|---|---|---|---|
| Gem$^{FL}$ | 1.3 ± 0.7 | 1.9 ± 1.4 | 12 ± 3 |
| Gem$^{76-145}$ | 5.9 ± 0.2 | 3.4 ± 0.6 | 5.7 ± 0.7 |
| Gem$^{76-140}$ | 3.3 ± 0.6 | 3.3 ± 0.4 | 11 ± 4 |
| Gem$^{76-135}$ | 6 ± 2 | 2.9 ± 0.2 | 7 ± 3 |

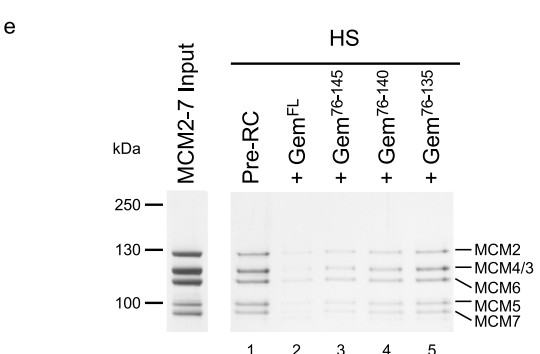

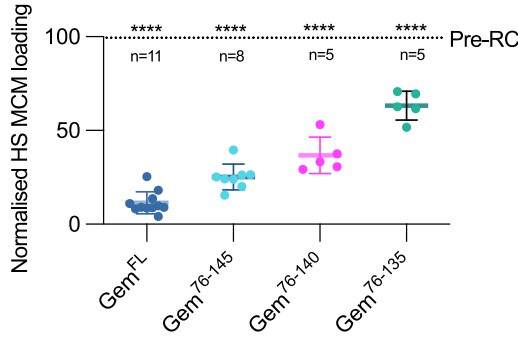

**Fig. 7 | Investigating C-terminal truncations of the geminin coiled-coil.**
**a** Alignment of the AlphaFold prediction of CDT1 (white) in complex with geminin (shown in cartoon form with tubular helices to highlight the C-terminus of the coiled-coil), with the human OCCM structure (PDB ID 8RWV). When aligning CDT1 in the AlphaFold structure with CDT1 in the human OCCM the steric clash between geminin and MCM2 (peach, shown in transparent surface view) is evident. Truncating the C-terminus of the geminin mimetic peptide reduces this clash. Figure made using Chimera X. **b** Cartoon of geminin peptides Gem$^{76-145}$, Gem$^{76-140}$, or Gem$^{76-135}$. Created in BioRender. Faull, S. (2025) https://BioRender.com/3k0eyee.

**c** Kinetic and dissociation constants for SPR experiments, n = 3, shown as mean ± SE. **d** Plot of $K_D$ values for binding of geminin peptides and immobilised CDT1$^{\Delta N}$ from SPR experiments. Data shown as the mean of 3 independent repeats ±SE. **e** Pre-RC HS-washed elutions after addition of Gem$^{FL}$, Gem$^{76-145}$, Gem$^{76-140}$ or Gem$^{76-135}$ to the pre-RC reaction. **f** Quantification of the mean MCM2-7 band intensity for reactions shown in (**e**) normalised to the control pre-RC reaction and compared mixed-effects analysis with Tukey's multiple comparisons test. ****$p < 0.0001$. Data shown as the mean ± SD. Source data are provided as a Source Data file.

complete CDK inhibition. CDC6 exhibited the strongest phosphorylation signal, approximately 100-fold greater than any other protein. Using CDK2/cyclin A, we observed similar results, although CDK2 did not phosphorylate CDT1 and MCM2-7.

Consequently, we asked what the impact of the phosphorylation was on pre-RC formation. Therefore, we added CDK2/cyclin A and CDK1/cyclin A2 into the pre-RC assay. The addition of either CDK led to a reduction in high-salt stable MCM2-7 loading (Fig. 8a, lanes 3 and 5).

Quantification revealed ~30% of activity remained in the presence of either CDK (Fig. 8b). When we combined CDK and geminin, no MCM2-7 loading could be detected (Fig. 8a, lanes 4 and 6). Thus, both geminin and CDK are required to fully inhibit DNA replication licensing.

To address the functional target protein of CDK1, we individually phosphorylated ORC1-5, ORC6, CDC6, CDT1$^{\Delta N}$ and MCM2-7 and then stopped the reaction by the addition of the CDK inhibitor p27 (Fig. 8c). A control reaction where all proteins were phosphorylated showed a reduction in MCM2-7 loading, when compared to an identical reaction that included the CDK inhibitor prior to pre-phosphorylation of licensing proteins. Thus, this experiment and our kinase assays with individual proteins highlight that the inhibitor is working (Fig. 8c, d and Supplementary Fig. 21). We observed that phosphorylation of ORC1-5 and CDC6 individually led to a reduction in MCM2-7 loading, while phosphorylation of both ORC1-5 and CDC6 resulted in a similar level of MCM2-7 loading inhibition as the reaction where only CDC6 was phosphorylated. On the other hand, prephosphorylation of ORC6, CDT1$^{\Delta N}$ or MCM2-7 had no impact on the reaction, despite CDT1$^{\Delta N}$ or MCM2-7 being phosphorylated by CDK1/cyclin A2 (Supplementary Fig. 21d, e). Thus, we conclude that ORC1-5 and CDC6 are the specific targets of CDK1/cyclin A2, resulting in a ~40% reduction of MCM2-7 loading (Fig. 8d). Similarly, pre-phosphorylation of only ORC1-5 and CDC6, along with the addition of geminin into the pre-RC assay, was sufficient to inhibit salt-stable MCM2-7 loading to near undetectable levels (Fig. 8e, f). Notably, a dual CDK/geminin requirement for inhibition ensures DNA licensing in G1-phase, when geminin is absent from chromatin and CDK2 and CDK1 are inactive and would block DNA licensing in S-, G2- and M- phase, when both are present (Fig. 8g). Moreover, this means that low geminin concentrations in late G1-phase are insufficient to block pre-RC formation. However, additional factors are likely acting in conjunction with CDK and geminin to block re-licensing of DNA outside of G1-phase[27], e.g. the ubiquitin-mediated destruction of ORC1 and CDT1 in S-phase[54,55].

## Discussion

Previous work had demonstrated that geminin inhibits DNA replication licensing. However, the precise mechanism of this inhibition has remained unknown. Experiments using the *Xenopus* cell-free system had shown that a CDT1-geminin complex is permissive for pre-RC formation in the context of a CDT1-depleted extract, while the addition of geminin to a *Xenopus* egg extract would block helicase loading and DNA replication[39]. This could be for several reasons. For one, this could be due to a different structural organisation of the CDT1-geminin complex when it exists in a 1:2 or higher order organisation[39,40]. Alternatively, increased protein concentrations may support the formation of a higher-order inhibitory CDT1-geminin hetero-hexamer[40]. It could also be that the *Xenopus* egg extract contains a geminin-quenching activity that drops geminin levels below a threshold that is capable of blocking pre-RC formation[56]. This is a really difficult question to address in a system that contains thousands of proteins. Thus, the establishment of a reconstituted human DNA replication licensing assay opened the door to its analysis. Our work reveals that human geminin and CDT1 do not form a higher-order complex larger than the 1:2 hetero-trimer, even at elevated concentrations. Instead, we could prove that the CDT1-geminin hetero-trimer is the active unit in inhibiting pre-RC formation. Finally, a preformed human CDT1$^{FL}$-geminin complex was as active in blocking helicase loading as the addition of geminin into a pre-RC reaction. Thus, our data demonstrate that the CDT1-geminin trimer is the functional unit to regulate DNA replication licensing and is consistent with the *Xenopus* egg extract containing an uncharacterised geminin-quenching activity.

To better understand geminin's role in regulating pre-RC formation, we have fine-mapped geminin's functions to its protein sequence (Fig. 8h). We combined the measurement of CDT1-geminin binding constants with the analysis of mutants and mimetic peptides in a fully

reconstituted DNA replication licensing assay. Our work was guided by the AlphaFold prediction of the CDT1-geminin structure to explore the role of previously structurally unresolved sections. As such, we could link physical interaction data with structural information and a functional assay, which has not been possible before. We identified that Gem$^{96-135}$ represents the minimal CDT1-binding region within the coiled-coil. Specifically, geminin residues 131–135 were essential for geminin dimerisation and CDT1 binding. This analysis is consistent with previous work that combined CDT1-geminin interaction data with cellular analysis[34,57]. In summary, the geminin coiled-coil represents a key CDT1 interaction domain.

AlphaFold generated specific insights into the N-terminus of geminin (residues 76–90). This region folds into a short helix, followed by a loop that interacts with CDT1. This represents a second site in geminin that is required for CDT1 binding. Although the entire region has not been crystalised, likely due to flexibility, our analysis is nonetheless consistent with previous work[34,57]. Without the helix and short loop, geminin peptides can bind to CDT1, but the complex is permissive in DNA replication licencing assays. The mutation of CDT1 amino acids that participate in the binding of geminin residues 76–90 completely blocked the inhibition of pre-RC formation. Moreover, we probed the role of geminin residues 76–90 by insertion of a flexible linker at the interface of the coiled-coil and the helix-loop region. The mutant displayed reduced inhibition of pre-RC formation. Interestingly, an AlphaFold prediction could map a previously published mutant to the interface of the coiled-coil and the short loop region (Supplementary Fig. 22). This mutant had no impact on the CDT1-geminin interaction, but was also permissive for helicase loading[58]. Similarly, a mouse CDT1 double mutant that affects the interaction with the coiled-coil and the loop also leads to a loss of inhibition[59]. In conclusion, removal of the geminin short loop/helix or altering their structure by mutations does not affect the CDT1-geminin interaction, but leads to inhibition failure, while mutation of the same interface on the CDT1 side also leads to a loss of inhibition. Moreover, insertion of a flexible linker between the loop and the coiled-coil leads to a reduction of inhibition. Thus, the data strongly suggest that both of geminin's CDT1 binding sites act in coordination and hint that this coordination could be necessary for the positioning of the coiled-coil of geminin.

It was surprising to see that the geminin and MCM2 binding sites did not overlap, excluding the most obvious inhibitory mechanism. Instead, we discovered that geminin acts by inducing a steric clash with MCM2, which stops pre-RC formation. Accordingly, truncation mutants that shortened the coiled-coil of geminin to a length that did not clash with MCM2, relieved the inhibitory activity of geminin. On the other hand, insertion of a linker into the geminin coiled-coil, which is predicted to maintain the overall coiled-coil shape, but modifies the coiled-coil relative orientation, had full inhibitory activity. Moreover, the sequence alignment of the geminin coiled-coil highlighted that the length of the coiled-coil is relatively well conserved and amino-acid identity is highest at the coiled-coil interface (Supplementary Fig. 17c, d). Thus, the data argue that the overall length and shape of the coiled-coil are important, but not its sequence beyond what is necessary to maintain the coiled-coil. In summary, we suggest that geminin inhibits DNA replication licensing by forming a tight CDT1-geminin complex with a coiled-coil that generates a steric clash with MCM2. What could be the advantage of this regulation? We suggest that independent geminin and MCM2 binding sites in CDT1 may allow for independent regulation of both interactions.

We observed that geminin did not completely inhibit DNA licensing but only reduced the activity by about 85% (Fig. 1d, e). Consequently, our data suggest that geminin must act with additional factors to inhibit DNA licensing in S-, G2- and M-phases. We identified CDK1/cyclin A2 and CDK2/cyclin A as factors that work together with geminin to inhibit DNA licensing (Fig. 8). This is an appealing concept, as it would restrict DNA licensing inhibition to S-, G2- and M- phases.

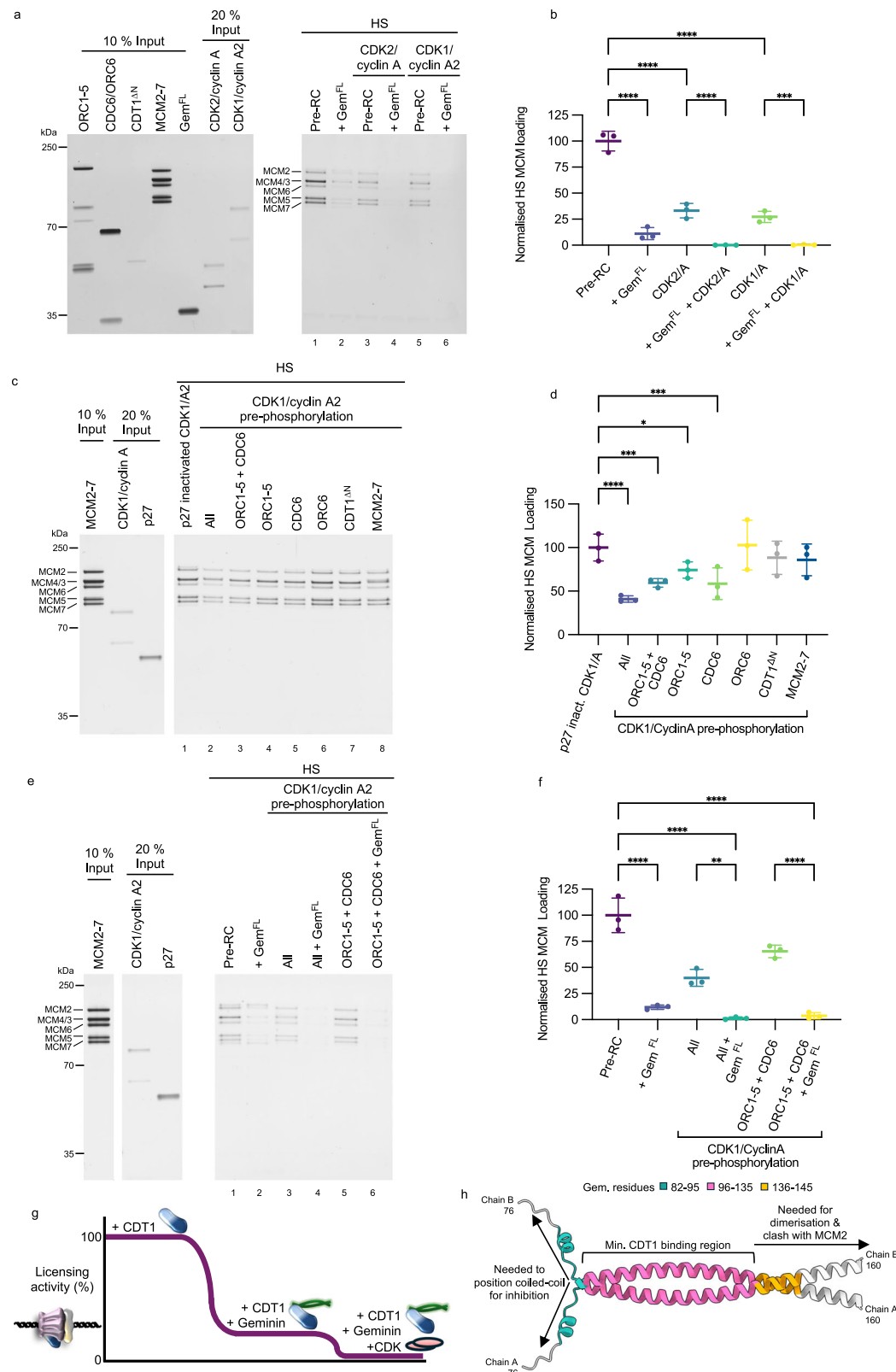

Although CDT1 has been recently linked to CDK1-specific control of rereplication[60], we identified in our system that ORC1-5 and CDC6 are the primary targets of CDK-dependent DNA licensing inhibition. Why this difference? CDT1 likely depends on accessory factors like PCNA or others for optimal phosphorylation[23,61], which are absent in our reconstituted system. Moreover, we note that we employed a CDT1 construct missing the N-terminus, which is missing key

phosphorylation sites[61]. Nevertheless, ORC1 has been firmly linked to CDK-dependent phosphorylation in yeast and humans[53,54,62]. Moreover, in yeast, CDK is the primary regulator of DNA licensing in S-phase[62]. Thus, we suggest that in humans, geminin works together with CDK to inhibit DNA licensing. Interestingly, in *Xenopus*, inhibition of CDKs is not sufficient to induce re-replication of DNA. However, in a geminin-depleted metaphase *Xenopus* extract, CDK inhibition leads to

**Fig. 8 | How does geminin contribute to the regulation of DNA replication licensing? a** Pre-RC assay using a combination of geminin and CDK/cyclin to assess effects on salt-stable MCM2-7 loading. **b** Quantification of high-salt stable loading from part (**a**). Data represented as the mean of three independent experiments ±SD. Relevant comparisons of conditions with and without geminin have been analysed using one-way ANOVA with Tukey's multiple comparisons test. ****P < 0.0001, ***P = 0.0003. **c** Pre-RC assay testing the impact of pre-phosphorylation of pre-RC protein components with CDK1/cyclin A2 on salt-stable MCM2-7 loading. **d** Quantification of high-salt stable loading form part (**c**). Data represented as the mean of three independent experiments ±SD. Data were analysed using one-way ANOVA with Dunnett's multiple comparisons test, ****p < 0.0001, ***p = from bottom to top; 0.0004, 0.0003, *p = 0.0167. **e** Pre-RC assay using a combination of pre-phosphorylation of pre-RC protein components with CDK1/cyclin A2 and the addition of geminin to assess the effects on salt-stable MCM2-7 loading.

**f** Quantification of high-salt stable loading form part (**e**). Data represented as the mean of three independent experiments ±SD. Data were analysed using one-way ANOVA with Tukey's multiple comparisons test, ****p < 0.0001, **p = 0.0024. **g** Graphical summary of pre-RC formation. In the presence of CDT1, DNA licencing can occur. The addition of geminin greatly inhibits DNA licensing activity, therefore co-operation between geminin and CDK is required for tight regulation. **h** Functional regions of geminin determined using mimetic peptides. The minimal CDT1 binding region (residues 96–135) are shown in pink. Residues 82–95 at the N-terminus of the peptide required to inhibit MCM2-7 loading through positioning of the coiled-coil are shown in teal and inhibitory residues at the C-terminus (136–145), needed for geminin dimerisation and to produce a steric clash with MCM2 in the OCCM, are shown in orange. Figure made using Chimera X. Source data are provided as a Source Data file.

greater re-replication, highlighting that CDK can also act in *Xenopus* as a DNA licensing inhibitor[48]. In Meier Gorlin disease, specific mutations affect the ORC1-cyclin A interaction, and it has been suggested these mutations may impact control of DNA licensing[52]. Moreover, the overexpression of cyclin E and A is linked to cancer and misregulated pre-RC formation[63–65], suggesting functional relevance of CDK regulation of DNA licensing in cells.

In budding yeast, geminin does not exist. However, budding yeast Cdt1 has a much larger MHD. Structural alignment of yCdt1 with human CDT1 and geminin (Supplementary Fig. 18c) revealed a clash between geminin and the larger yMHD. Thus, this provides an explanation, why the human CDT1 is more compact, missing the extra domain and freeing-up space for geminin to attach to CDT1. Geminin has previously been explored as a DNA replication licensing inhibitor through overexpression of a non-degradable form that was generated by the mutation of the d-box to avoid APC/C-dependent degradation during M-phase[66]. Overexpression of this construct resulted in the apoptosis of cancer cells, whilst arresting non-cancerous cells. In *Xenopus* egg extract assays, a Gem[76–145] peptide, which also lacks the d-box was able to inhibit DNA replication licensing efficiently. Although these peptides are too long for chemical synthesis, the identified interaction surfaces could represent targets for drug development. Since geminin-mediated inhibition is not complete, we suggest that these inhibitors could be combined with a CDT1 CHD-targeting activity to inhibit DNA licensing in G1-phase completely. Thus, the knowledge we gained may help to design better DNA replication inhibitors.

Thus, our data demonstrate a dual CDK and geminin inhibitory mechanism. We like this model, as it would make DNA licensing insensitive to low levels of CDK or geminin activity in late G1-phase. Thus, inhibitory activity can be built up ahead of time, facilitating a more rapid shut-off in S-phase. Importantly, in cells, additional pathways outside of the regulation of CDT1 activity and stability exist[27]. Moreover, the reconstituted system currently has limitations, including the absence of post-translational modifications and chromatin modifications, as the proteins were purified from bacteria, and the DNA was not covered by histones. These additional layers of regulation can regulate ORC1 protein stability in S-phase[54], or access to the DNA[67], which in turn can block DNA replication licensing. Thus, although geminin and CDK appear sufficient to block DNA licensing completely, additional pathways can regulate specific loci or can enhance the quality of the overall regulation.

## Methods
### Expression and purification of ORC1-5
ORC1-5 was expressed and purified as described[10]. Expression plasmids were designed based on pESC vectors (Stratagene) and generated by Genscript (Supplementary Data 1). The three plasmids were co-expressed[6] in *S. cerevisiae* YC658 cells (MATa, lys2::pGAL1 GAL4::-LYS2, pep4::HIS3, bar1::hisG derived from W303)[68]. Cells were lysed

using a SPEX freezer-mill and resuspended in lysis buffer (50 mM HEPES, 250 mM NaCl, 5% (v/v) glycerol, 2 mM DTT, 0.02% (v/v) NP-40, 2 mM MgCl$_2$ 50 mM NaF, 10 mM β-glycerol phosphate, 1 mM ATP; pH adjusted to 7.8), containing benzonase (Merck), complete EDTA-free protease inhibitor tablets (Roche) and PhosSTOP phosphatase inhibitor cocktail tablets (Roche). Lysates were clarified before loading onto a StrepXT column (Cytiva). The column was washed with lysis buffer, and protein eluted by addition of 75 mM biotin. 2 mM MgCl$_2$ and 1 mM ATP were added to stabilise eluted protein prior to dilution to 200 mM NaCl. For some purifications, the N-terminal Strep-tag on ORC1 was cleaved overnight using SUMOstar protease (LifeSensors), the activity of tagged and untagged ORC1-5 has previously been shown to be equivalent[10]. Pooled fractions were bound to a 1 mL HiTrap Heparin column (Cytiva) equilibrated in low salt buffer (50 mM HEPES-NaOH (pH 7.6), 200 mM NaCl, 1.5 mM DTT, 25 mM NaF, 5% (v/v) glycerol, 2 mM MgCl$_2$, 0.02% (v/v) NP-40), before elution by a salt gradient using 0.2–1 M NaCl.

### Expression and purification of ORC6
ORC6 was expressed and purified as described[10]. The expression plasmid was manufactured by Genscript using pET21a as a backbone (Supplementary Data 1). His-ORC6 was expressed in *Escherichia coli* (*E. coli*) Rosetta 1 (DE3) cells (Agilent) for 16 h at 16 °C. Harvested cells were resuspend in lysis buffer (50 mM HEPES (pH 7.5), 150 mM NaCl, 10 mM Imidazole, 5% (v/v) glycerol, 1 mM DTT) and sonicated on ice. Lysates were clarified by centrifugation prior to loading onto a HisTrap Excel column (Cytiva) that has been equilibrated with lysis buffer. Protein was eluted using a gradient of up to 450 mM imidazole. The His-tag was removed by overnight cleavage with PreScission protease (Cytiva) at 4 °C. Untagged protein was bound to a POROS HQ column in buffer A (30 mM HEPES-NaOH (pH 7.5), 150 mM NaCl, 1 mM DTT). Protein was eluted using a gradient of up to 70% buffer B (30 mM HEPES-NaOH (pH 7.5), 1 M NaCl, 1 mM DTT). Fractions were then loaded onto a HiLoad Superdex 75 16/60 column equilibrated in buffer C (10 mM HEPES-NaOH (pH 7.5), 150 mM NaCl, 1 mM DTT).

### Expression and purification of CDC6
CDC6 was expressed and purified as described[10]. The CDC6 expression plasmid was generated by Genscript using pGEX6P1 (Cytiva) as a backbone (Supplementary Data 1). CDC6 was expressed overnight at 16 °C in *E. coli* BL21 codon + RIL cells (Agilent). Cells were sonicated in lysis buffer (50 mM HEPES-NaOH (pH 7.6), 250 mM KCl, 50 mM NaCl, 2 mM MgCl$_2$, 0.02% (v/v) NP-40, 10% (v/v) glycerol, 2 mM DTT). Clarified lysate was bound to GST-Agarose resin (Sigma) for 1 h at 4 °C on a rotator before washing with 10 column volumes (CV) lysis buffer supplemented with 1 mM ATP. Resin was further washed with 10 CV high salt buffer (50 mM HEPES-NaOH (pH 7.6), 1 M KCl, 50 mM NaCl, 2 mM MgCl$_2$, 0.02% (v/v) NP-40, 10% (v/v) glycerol, 1 mM DTT, 1 mM ATP), and 3 CV buffer C (30 mM HEPES-NaOH (pH 7.6), 167 mM KCl, 33.2 mM NaCl, 1 mM MgCl$_2$, 0.02% (v/v) NP-40, 5% (v/v) glycerol, 1 mM

DTT, 1 mM ATP). 2 CV of buffer C containing PreScission protease (Cytiva) was then added to the resin for overnight cleavage at 4 °C. The following morning, the protein that had been eluted from the resin was loaded onto a HiTrap SP HP column (Cytiva) equilibrated in buffer C. A gradient elution was performed by adding increasing amounts of buffer D (30 mM HEPES-NaOH (pH 7.6), 825 mM KCl, 165 mM NaCl, 1 mM DTT, 5% (v/v) glycerol, 1 mM MgCl$_2$, 0.05% (v/v) NP-40) to 100%.

### Expression and purification of CDT1

CDT1$^{FL}$ and the N-terminally truncated CDT1$^{158-546}$ (referred to as CDT1$^{\Delta N}$) were expressed and purified as described[10]. Plasmids were generated by Genscript (Supplementary Data 1). CDT1$^{FL}$ used pGEX6P1 (Cytiva) as a backbone to produce a construct with an N-terminal GST-tag. The plasmid was transformed into E. coli BL21 (Agilent) and protein expressed overnight at 16 °C. Cell pellets were sonicated in 50 mM HEPES-NaOH (pH 7.6), 250 mM NaCl, 2 mM DTT, 0.1% (v/v) Triton X-100, 10% (v/v) glycerol. Clarified lysate was incubated with Sepharose Glutathione FastFlow resin (Sigma) at 4 °C for 2 h. The GST-tag was removed by the addition of PreScission protease (Cytiva) overnight at 4 °C. Buffer B (30 mM HEPES-NaOH (pH 7.6), 200 mM NaCl, 1 mM DTT, 0.01% (v/v) Triton X-100, 5% (v/v) glycerol) was used to wash protein from the column. Collected protein was loaded onto a POROS™ HS 20 µm column (ThermoFisher) and eluted using a 0.2 to 1 M NaCl gradient. Further purification was then performed using a Superdex 200 Increase 10/300 GL column (Cytiva) equilibrated in 10 mM HEPES-NaOH (pH 7.6), 200 mM NaCl, 1 mM DTT, 0.01% (v/v) Triton X-100, 5% (v/v) glycerol.

The N-terminally truncated CDT1$^{158-546}$ (referred to as CDT1$^{\Delta N}$) was cloned into pET21A (+) to produce a construct with a 6xHis tag at the N-terminus. The plasmid was transformed into BL21 (DE3) (Agilent) and expressed overnight at 16 °C. Cells were lysed via sonication in 50 mM HEPES-NaOH (pH 7.6), 200 mM NaCl, 10 mM imidazole, 2 mM DTT, 0.02% (v/v) NP-40, 10% (v/v) glycerol. The lysate was clarified and then loaded onto a HisTrap Excel column (Cytiva) before elution with 10 to 450 mM imidazole. Dialysis was performed overnight into 30 mM HEPES-NaOH (pH 7.6), 200 mM NaCl, 1 mM DTT, 0.02% (v/v) NP-40, 5% (v/v) glycerol. Protein was loaded onto a POROS™ HS 20 µm column (ThermoFisher) and eluted with a gradient of 200–1000 mM NaCl. Selected fractions were then loaded onto a Superdex 200 Increase 10/300 GL column (Cytiva) equilibrated with 10 mM HEPES-NaOH (pH 7.6), 200 mM NaCl, 1 mM DTT, 0.02% (v/v) NP-40, 5% (v/v) glycerol. For the CDT1 MHD (CDT1$^{158-396}$) and CHD (CDT1$^{391-546}$) purifications, the POROS™ HS step was omitted. The CDT1$^{4A}$ and CDT1$^{R330A}$ mutants were generated by Genscript using the CDT1$^{\Delta N}$ plasmid as a parent vector and purified as for the MHD and CHD mutants.

For Cy3-labelled CDT1$^{\Delta N}$, a plasmid containing ybbR-CDT1$^{\Delta N}$ was generated by Genscript using a pGEX6P1 (Cytiva) backbone (Supplementary Data 1). Protein was produced overnight at 16 °C in E. coli BL21 (Agilent) cells. Pellets were resuspended in lysis buffer (50 mM HEPES, 250 mM NaCl, 10% (v/v) glycerol, 2 mM DTT, 0.1% (v/v) Triton X-100, pH 7.6) supplemented with SIGMAFAST™ protease inhibitor EDTA-free cocktail tablets (Sigma) and benzonase (final concentration 10 U/mL, Sigma) and sonicated. Clarified lysate was incubated with Sepharose Glutathione Fast Flow beads (Sigma) for 2 h at 4 °C. Beads were washed with lysis buffer, followed by high salt buffer (50 mM HEPES, 1 M NaCl, 10% (v/v) glycerol, 2 mM DTT, 0.1% (v/v) Triton X-100, pH 7.6). Cleavage buffer (30 mM HEPES, 200 mM NaCl, 5% (v/v) glycerol, 1 mM DTT, 0.01% (v/v) Triton X-100, pH 7.6 containing 5 µg mL$^{-1}$ homemade GST-tagged HRV3C protease) was then added overnight with mixing at 4 °C. Protein was labelled using the protocol described[69]. ybbR-CDT1 was concentrated and then supplemented with 10 mM MgCl$_2$, 76 µM Cy3-Coenzyme A and 1 µM Sfp phosphopantetheinyl transferase. Labelling was performed overnight at 4 °C. Separation of Cy3-labelled protein was achieved using a Superdex 200 Increase 10/300 GL column (Cytiva) equilibrated in

10 mM HEPES, 200 mM NaCl, 5% (v/v) glycerol, 1 mM DTT, 0.01% (v/v) Triton X-100, pH 7.6.

### Expression and purification of MCM2-7

MCM2-7 was expressed in HEK293-F cells by Oxford Expression Technologies Ltd (Oxford, UK) as previously described[10,70] using the plasmids described in Supplementary Data 1. Pellets were resuspended in lysis buffer (20 mM HEPES-NaOH (pH 7.5), 250 mM KGlu, 5 mM MgCl$_2$, 2 mM ATP, 10 mM imidazole, 0.02% (v/v) NP-40, 5% (v/v) glycerol) containing SIGMAFAST™ EDTA-free protease inhibitor tablet and Benzonase prior to sonication. Centrifugation (41,000 x $g$ for 50 min) was used to remove the membrane fraction. The supernatant was loaded onto Ni-NTA agarose (Qiagen). Protein was eluted using 300 mM imidazole prior to loading onto a StrepTrapXT column (Cytiva) equilibrated with 20 mM HEPES-NaOH (pH 7.5), 250 mM KGlu, 5 mM MgCl$_2$, 2 mM ATP, 1 mM DTT, 0.02% (v/v) NP-40, 5% (v/v) glycerol. Fractions were eluted using 50 mM biotin.

### Purification of Gem$^{FL}$ (geminin), recGem$^{76-145}$, Gem$^{LINK}$ and Gem$^{2xGGS}$

DNA fragments were cloned into pET21A (+) plasmids by Genscript to produce constructs with an N-terminal 6xHis tag (Supplementary Data 1). Plasmids were expressed overnight at 16 °C in E. coli BL21 (DE3) (Agilent) cells. Pellets were resuspended in 50 mM HEPES, 150 mM NaCl, 5% (v/v) glycerol, 1 mM DTT, 10 mM imidazole, pH 7.5 containing SIGMAFAST™ protease inhibitor EDTA-free cocktail tablets and 10 U/mL benzonase (both Sigma) and sonicated. Clarified lysate was applied to a HisTrap Excel column, and protein was eluted using a gradient of 10–450 mM imidazole. The His-tag was cleaved using 5 µg/mL homemade GST-tagged HRV3C protease whilst dialysing into 30 mM HEPES, 150 mM NaCl, 5% (v/v) glycerol, 1 mM DTT, pH 7.5 overnight at 4 °C. The protein was when loaded onto a POROS™ HQ 20 µm column (Thermo Scientific) and eluted with 150–1000 mM NaCl, except for Gem$^{2xGGS}$ where this step was omitted. Finally, the protein was a subjected to size exclusion chromatography (Superdex 200 Increase 10/300 GL column, Cytiva) in 10 mM HEPES, 150 mM NaCl, 5% (v/v) glycerol, 1 mM DTT, pH 7.5.

### Purification of CDT1-geminin (CG) complex

Full-length geminin (N-terminal 6xHis-tag) and CDT1 (untagged) (see Supplementary Data 1 for plasmids) were co-expressed in E. coli BL21 (DE3) (Agilent) overnight at 16 °C. The purification was then performed as described for geminin.

### Human pre-RC assay

The assay was performed as described[10]. 47 nM ORC1-5, 94 nM ORC6, 94 nM CDC6, 47 nM CDT1 and 70 nM MCM2-7 were incubated in pre-RC buffer (25 mM HEPES-KOH (pH 7.5), 250 mM KGlu, 1 mM DTT, 4 mM MgOAc, 0.1% (v/v) Triton X-100, 1 mM ATP or ATPγS) for 10 min at 30 °C with shaking at 300 RPM in a ThermoMixer C (Eppendorf) to allow for nucleotide binding. Where applicable, geminin was added at 94 nM and mimetic peptides were typically used at 50x excess (4.7 µM). 600 ng of human B2-lamin dsDNA, amplified from plasmid pCS1087 as either a 2 kbp or 3 kbp fragment (see Supplementary Data 1 for oligo sequences), conjugated to magnetic streptavidin beads (MyOne Streptavidin T1, Invitrogen) was then added and the mixture was incubated for a further 20 min. Beads were washed twice with either a low- (pre-RC buffer) or a high- salt (pre-RC buffer with 300 mM NaCl) buffer. Endonuclease DNaseI (Thermo Scientific) was used to elute complexes from beads using pre-RC buffer plus 5 mM CaCl$_2$. Proteins were typically resolved using SDS-PAGE and visualised using silver staining.

For pre-RC assays using CDK, the reaction was set-up in the normal way with the addition of 23.5 nM of the relevant CDK/cyclin (CDK1/cyclin A2 (#PV6280) or CDK2/cyclin A (#PV3267), both Thermo Fisher)

added to ORC1-5, CDC6, ORC6, CDT1$^{\Delta N}$ and MCM2-7 for 5 min, 30 °C, 300 RPM. After this phosphorylation step, geminin was added to the relevant reactions for 10 min before the addition of DNA-conjugated magnetic beads. When phosphorylating individual protein components, pre-RC proteins were incubated with 23.5 nM CDK1/cyclin A2 for 30 min at 30 °C, 300 RPM. P27KIP1 (SRP5109, Sigma) was added at a 2× molar excess to CDK1/cyclin A2 and incubated for a further 5 min to inhibit CDK1/cyclin A2 kinase activity. Then, the relevant non-phosphorylated proteins were then added to the reaction mixture along with DNA-conjugated magnetic beads to facilitate licensing formation.

For the two-step pre-RC assay, reactions were assembled as described in ATPγS containing pre-RC buffer and using Cy3-CDT1$^{\Delta N}$. Eighteen minutes after the addition of DNA, the control reactions were washed twice with low-salt buffer (ATPγS) before elution. Geminin was spiked into the remaining reactions for 1, 2 or 4 min prior to washing and eluting.

### Western blotting

For the anti-CDT1 blots, pre-RC reactions were transferred onto PVDF membrane and blocked with milk in TBS-T prior to overnight incubation with an anti-CDT1 antibody (F-6, raised against residues 247-546, sc-365305, lot L2821, Santa Cruz, 1:1000). The signal was detected using an HRP-conjugated anti-mouse secondary antibody (Sigma A4416, 1:10000). CDT1 band intensity was measured using Multi Gauge V2.3 (FujiFilm) software. The signal for reactions were normalised to the CDT1 loading control. For the kinase assays, blotting was performed as described above, but with Phospho-CDK Substrate Motif [(K/H)pSP] primary antibody (Cell Signalling Technology, #9477 lot 2, 1:1000) and anti-rabbit IgG HRP-linked secondary antibody (Cell Signalling Technology, #7074, 1:10000).

### Kinase assays

For the CDK experiments, reactions were assembled in 50 μl pre-RC buffer. 23.5 nM CDK/cyclin and 47 nM of p27 were added to relevant tubes for 5 min, 30 °C, 300 RPM before the addition of 47 nM ORC1-5, 94 nM ORC6, 94 nM CDC6, 47 nM CDT1 and 70 nM MCM2-7 (same concentrations as used in the pre-RC assay) for a further 30 min. The reaction was stopped by the addition of 12.5 μl 4× Laemmli buffer. For the CDK-p27 control, ORC1-5, ORC6 and CDT1 25 μl was loaded. For CDC6 the equivalent of 0.25 μl was loaded and for MCM2-7 4 μl was loaded to make the signal comparable in western blots.

### Gel quantification and statistics

Silver-stained SDS-PAGE gels were quantified using Multi Gauge V2.3 (FujiFilm) software. The density of each MCM2-7 protein band was measured relative to the background of the gel and summed to determine the MCM2-7 signal for a given lane. Then, the MCM2-7 signal was normalised relative to a control reaction to determine relative MCM2-7 loading. Replicates represent results calculated across multiple experiments and have been pooled for each condition. For the CDT1 CHD and MHD combination experiment, high-salt loading was assessed using MCM3 and MCM4 bands only due to low signal-to-noise. For CDC6, the signal of the protein band was measured relative to the background of the gel. For fluorescent protein gels, image analysis was performed using a FujiFilm FLA-5100 Fluorescent Image Analyser and Multi Gauge V2.3 (FujiFilm) software. Cy3-fluorescence was visualised using a 532 nm laser. Data are displayed as the mean and standard deviation (SD), determined using at least three biological replicates. Statistical analysis was performed in GraphPad Prism 10 using recommended tests based on the dataset. One-way ANOVA was used in most cases, but for datasets with less than two groups, the two-tailed paired t-test was used. Datasets with missing values were analysed using mixed effects analysis. **** denotes p values < 0.001 and

not significant (ns) denotes a p value > 0.05. P values that fall between this range are stated.

### Negative stain electron microscopy

Purified MCM2-7 (1.56 μM) was pre-incubated at 30 °C, shaking at 600 RPM (in MCM2-7 purification buffer). A mixture of purified ORC1-5 (108 nM), ORC6, CDC6, CDT1$^{\Delta N}$ and geminin (all 216 nM) was incubated in pre-RC buffer (25 mM HEPES-NaOH (pH 7.5), 250 mM KGlu, 1 mM DTT, 4 mM MgOAc, 0.05% (v/v) Triton X-100, 1 mM ATP) in the same way in parallel. 480 nM 158 bp biotinylated lamin B2 dsDNA was incubated with 2 molar equivalents of streptavidin in pre-RC buffer under the same conditions. After 10 min, an aliquot of the DNA-streptavidin mixture was added to the reaction mixture containing ORC1-5 (now at 157.4 nM), ORC6, CDC6, CDT1$^{\Delta N}$ and geminin (now at 79.8 nM) so that the concentration of DNA was 130 nM and streptavidin was 260 nM. After a further 10 min of incubation, an aliquot of preincubated MCM2-7 was added to the reaction mixture. The final concentrations in this mixture were: ORC1-5, MCM2-7 (75 nM), ORC6, CDC6, CDT1$^{\Delta N}$, geminin (150 nM), B2 DNA (120 nM), streptavidin (240 nM). After 60 min of incubation, 3.5 μl of the reaction mixture was applied to a 300 mesh Cu grid, which had been glow discharged for 45 s using a PELCO easiGlow device. The grid was washed 2 times briefly with pre-RC buffer from which ATP and Triton X-100 were omitted. The grid was washed once with 2% uranyl acetate, then stained for a further 60 s with 2% uranyl acetate and blotted dry. For the control reactions, CDT1$^{\Delta N}$ or geminin were omitted as indicated. Data were collected at the MRC-LMS EM Facility using a Thermo Fisher Talos F200i TEM, equipped with a Falcon 3EC direct electron detector, at 200 kV. Micrographs were automatically collected using EPU software at ×73 k magnification using a defocus range from −1.0 to −3.0 μm and a pixel size of 2.0 Å/pixel. 500–1000 micrographs per condition were imported into in CryoSPARC (v4.6.2)[71], particles were picked using the blob picker (130–300 Å) and extracted with a box size of 512 Å. After two rounds of 2D classification, classes which could be unambiguously identified as double hexamers were selected and the number of double hexamers per micrograph was calculated from this.

### Mass Photometry

Measurements were acquired at room temperature (RT) using a TwoMP Mass Photometer (Refeyn Ltd). Purified proteins were incubated in the following ratios 0:1, 1:0, 1:2, 1:3, 2:4 and 1:4 (CDT1:geminin monomer, where 1 part is equal to a concentration of 150 nM) in pre-RC buffer minus Triton X-100. Samples were then diluted on the instrument 1:10 in PBS in a six-well silicone gasket (GraceBioLabs) on a pre-cleaned sample coverslip (Refeyn Ltd). For the MHD and CHD fragment experiments, proteins were incubated in a 1:2 ratio (CDT1:geminin monomer, where 1 part is equal to a concentration of 150 nM). AcquireMP software (v. 2023R2, Refeyn Ltd) was used to record 60 seconds movies in the regular field of view. Calibration was performed prior to recording samples using BSA (Sigma, 66 kDa monomer, 132 kDa dimer) and Bovine Thyroglobulin (Sigma, 670 kDa) in PBS to produce a linear mass calibration. The maximum mass error accepted for calibration was below 5%, as defined by the DiscoverMP software. Analysis was performed and figures generated using the DiscoverMP software (v2023R2, Refeyn Ltd). Measurements were completed in duplicate or triplicate, with one representative trace shown for each condition. Where the molecular weight was below the limit of detection (<30 kDa), histograms were not fitted to the data.

### SEC-MALS

Size exclusion chromatography was performed using an Agilent 1260 isocratic pump and variable wavelength detector connected to a Superdex 200 Increase 10/300 GL column (Cytiva). This apparatus was coupled with a miniDAWN TREOS MALS unit and an Optilab T-rEX dRI

detector (Wyatt Technology). Peptides (500–1000 μM, 100 μL) were dissolved in running buffer (10 mM HEPES, 150 mM NaCl, 1 mM TCEP-HCl at pH 7.5) and injected with a flow rate of 0.25 mL/min. An average refractive index increment (dn/dc) of 0.185 mL/g was detected using a laser wavelength of 657 nm, which was used to determine peptide/protein concentrations[72]. For each peptide, three biological repeats were measured, and the data was analysed using ASTRA 6 software (Wyatt Technology) and modelled using the Zimm model to determine mean and SE $M_W$ values[73].

## Solid phase peptide synthesis (SPPS)
Peptides were synthesised on a 0.050 mmol scale using a Liberty Blue™ Automated Microwave Peptide Synthesiser (CEM) using a typical 9-fluorenylmethyloxycarbonyl/tert-butyl (Fmoc/tBu) protection strategy. N,N-dimethylformamide (DMF) was used as the solvent and Tentagel S-RAM Rink Amide (0.24 mmol/g) or Rink Amide ProTide (0.19 mmol/g) resins were used. Within Fmoc-amino acid coupling steps, the Fmoc-protected amino acid (0.250 mmol, 5 eq, 0.2 M in DMF), Oxyma (0.250 mmol, 5 eq, 0.5 M in DMF), and N,N'-Diisopropylcarbodiimide (0.500 mmol, 10 eq, 0.5 M in DMF) were added to the resin and mixed at 90 °C for 2 min (single coupling), 2 × 2 min (double coupling), or 3 × 2 min (triple coupling). For coupling of Fmoc-histidine-OH amino acids, a modified protocol was used with a final temperature of 50 °C and a reaction time of 10 min. Fmoc-deprotection was performed by mixing the Fmoc-peptide chain with piperidine (10% (v/v) in DMF) and Oxyma (0.1 M in DMF) at 90 °C for 1.5 min. After automated SPPS, peptidyl resins were washed with DMF (5 × 5 mL) and dichloromethane (DCM) (5 × 5 mL) and were stored at 4 °C before N-terminal capping with acetic anhydride (Ac₂O).

## N-terminal acetylation of peptidyl-resins
To cap the peptide chain with Ac₂O, peptidyl resins (0.050 mmol) were swollen through the addition of DMF (3 mL) and 15 min of subsequent shaking at RT. The solvent was drained and a solution of N,N-Diisopropylethylamine (DIPEA) (5% (v/v)) and Ac₂O (5% (v/v)) in DMF (3 mL) was added. The mixture was shaken for 10 min at RT then the solvent was drained, and the peptidyl resin was washed with DMF (1 × 5 mL). This process was repeated again for two further 10 min cycles, then the peptidyl resin was washed with DMF (5 × 5 mL) and DCM (5 × 5 mL) and was stored at 4 °C prior to cleavage of the peptide from the resin.

## Cleavage of peptides from resin
A solution of trifluoroacetic acid (TFA) (2775 μL), H₂O (75 μL), triisopropylsilane (TIS) (75 μL), and 2,2′-(Ethylenedioxy)diethanethiol (DODT) (75 μL) was added to the N-acetylated peptidyl resin and the resulting mixture was shaken for 3 h at RT. The solvent was decanted and ice-cold diethyl ether (Et₂O) (30 mL) was added, resulting in the precipitation of a yellow-colourless solid. The mixture was cooled to −20 °C for 1 h and then was subjected to centrifugation (3400 × g, 3 min). The supernatant was decanted and this process was repeated thrice more using 3 × 15 mL of ice-cold Et₂O. The precipitate was dissolved in a solution of H₂O/MeCN (80:20, 5 mL) and lyophilised to yield the crude peptide as a yellow-colourless solid. Peptides were purified using high-performance liquid chromatography (HPLC) prior to use in biochemical and biophysical assays.

## Preparative HPLC
Preparative HPLC was performed using a LC-20AR preparative HPLC system (Shimadzu) equipped with an Aeris Peptide 5 μm XB-C18 column (Phenomenex) (150 mm × 21 mm, 5 μm, 100 Å). Crude peptide samples were dissolved in H₂O/MeCN (80:20, 12 mL), centrifuged at 21,000 × g for 2 min, and then filtered with 0.2 μm polytetrafluoroethylene (PTFE) syringe filters. A linear gradient of 20–50% of HPLC solvent A (MeCN + 0.08% TFA) in HPLC solvent B (H₂O + 0.1% TFA) was used over 20 min with a flow rate of 20 mL/min. Fractions

were identified using a detection wavelength of 220 nm and purities were determined using analytical HPLC. Fractions with purities >95% were pooled and lyophilised, yielding purified peptides as colourless solids (Supplementary Fig. 23).

## Analytical HPLC
Analytical HPLC was performed using a LC-2030C 3d Plus (Shimadzu) equipped with an Aeris Peptide 3.6 μm XB-C18 column (Phenomenex). A linear gradient of 20–95% HPLC solvent A in HPLC solvent B was applied over 15 or 24 min with a flow rate of 1.5 mL/min. Chromatograms were recorded at a wavelength of 220 nm (Supplementary Fig. 24a–p). Peaks were detected, peak areas integrated, and peptide purities determined using LabSolutions software (Shimadzu).

## High-resolution LC-MS of pure peptides
Molecular weights of purified peptides were determined using a 6545XT AdvanceBio LC/Q-TOF system (Agilent) equipped with an Aeris Peptide 3.6 μm XB-C18 column (Phenomenex). A linear gradient of 20–50% HPLC solvent C (MeCN + 0.1% formic acid) in HPLC solvent D (H₂O + 0.1% formic acid) was applied over 5 min with a flow rate of 0.4 mL/min. For each peptide, one LCMS sample was prepared by dissolving the lyophilised peptide powder in a solution of 20% HPLC solvent C in HPLC solvent D, with a final concentration of ~0.1 mg/mL. Each peptide solution was injected one time, and masses were detected in the 400–2000 m/z range. The mass spectra were deconvoluted and the molecular ion peak calculated using MassHunter Qualitative Analysis 10.0 software (Agilent). MassHunter isotope distributor calculator software (Agilent) was used to verify that observed masses were in agreement with calculated masses. Spectra are shown in Supplementary Fig. 25a-n.

## AlphaFold
The AlphaFold server[49] was used to perform predictions of protein/peptide complexes from provided amino acid sequences. For CDT1 Uniprot Q9H211 was used and for geminin O75496. Output structures were ranked according to prediction confidence per residue scores (predicted Local Distance Difference Test, pLDDT) and the highest-ranked structure was selected. Molecular graphics and analyses performed with UCSF ChimeraX-1.9, developed by the Resource for Biocomputing, Visualisation and Informatics at the University of California, San Francisco, with support from National Institutes of Health R01-GM129325 and the Office of Cyber Infrastructure and Computational Biology, National Institute of Allergy and Infectious Diseases. UCSF Chimera X-1.9 was used to display pIDDT scores. The PAE plot and corresponding models were downloaded from PAE viewer[74]. PAE scores for interaction regions were obtained using PAE Viewer[74].

## BUDE alanine scanning
The BAlaS[51] web server (https://pragmaticproteindesign.bio.ed.ac.uk/balas/.) was used to perform computational alanine scanning. Receptor (geminin chains A and B) and ligand (CDT1 chain C) proteins were designated from the AlphaFold output file and the Bristol University Docking Engine (BUDE) was used to compute ΔΔG values. BUDE uses a forcefield approach to estimate the free binding energy of the alanine substitution of each ligand residue. Results are displayed using the Chimera (1.17.3) output provided by the server.

## Protein sequence analysis
Protein sequence alignments were obtained using Clustal Omega and analysis was performed using ESPript and the ConSurf Server[75,76].

## SPR binding assays
Assays were performed using a Biacore S200 system (Cytiva) at 25 °C in 10 mM HEPES, 500 mM NaCl, 50 μM EDTA, 1 mM TCEP-HCl, 0.05%

(v/v) Tween-20, pH 8.0. The binding of Gem$^{FL}$ to CDT1 was investigated using single-cycle SPR experiments. CDT1$^{\Delta N}$ was covalently immobilised using a Series S sensor chip CM5 (Cytiva). A 7 min pulse (10 μL/min) of 0.2 M 1-Ethyl-3-diaminopropyl carbodiimide (EDC)/0.05 M N-hydroxysuccinimide (NHS) mixture was used to activate the chip surface. 10 nM CDT1$^{\Delta N}$ was immobilised by a 1 min pulse (5 μL/min) before a 7 min neutralisation pulse of 1 M ethanolamine (10 μL/min). Absolute response levels were typically increased by ~50 RU after injection. A total of 7 injections were used for single-cycle SPR experiments using a three-fold concentration range of from 0.91 pM to 750 nM of the Gem$^{FL}$ dimer. For both association and dissociation stages, the flow rate was 30 μL/min with a contact time of 150 s and a final dissociation time of 1000 s. Seven pulses of running buffer (45 s, 30 μL/min) were then used so further single-cycle experiments to be performed without the need to re-immobilise CDT1$^{\Delta N}$.

The binding of geminin mimetic peptides to CDT1 was observed using multi-cycle SPR experiments. CDT1$^{\Delta N}$ was immobilised using the interaction between its N-terminal His-tag and Ni$^{2+}$ on the surface of a Series S sensor chip NTA (Cytiva). A 60 s pulse of 0.5 mM NiCl$_2$ (10 μL/min) was used for surface activation, followed by a 1 min pulse of 40 nM CDT1$^{\Delta N}$ (5 μL/min) to immobilise the protein on the sensor surface. Absolute response levels typically increased by ~200 RU. Seven 45 s pulses of running buffer at a rate of 30 μL/min were performed before injection of the peptide solution (30 μL/min, 120 s contact time and 500 s dissociation time). 60 s pulses of 500 nM imidazole (30 μL/min), then 350 mM EDTA (30 μL/min), were used to regenerate the surface before NiCl$_2$ was used to immobilise CDT1$^{\Delta N}$ and the next peptide concentration was injected. A two-fold concentration range from 0.49–250 nM (10 concentrations total) was used for each geminin mimetic peptide. Kinetic binding analysis was not performed for any injections that caused substantial bulk shifts. For both single- and multi- cycle experiments Biacore Evaluation software (Cytiva) was used to analyse the double-referenced sensorgrams. The raw data was subtracted from a blank injection and reference surface responses. All analytes were modelled using a 1:1 kinetic binding model of analyte dimer:CDT1, for except for Gem$^{LINK}$, Gem$^{96-160}$ and Gem$^{102-160}$, which used a steady-state 1:1 binding model. The average $k_{on}$, $k_{off}$ and $K_D$ values of at least three biological repeats are reported, alongside standard error (SE) values.

## CD spectroscopy

CD spectroscopy was performed using a Chirascan V100 CD spectrometer (Applied Photophysics) and a Hellma QS quartz cuvette with a 1 mm pathlength. Peptides were dissolved in CD buffer (10 mM NaH$_2$PO$_4$, pH 7.4) with a final concentration of 50 μM. CD measurements were obtained at 25 °C using a wavelength range of 190–260 nm with a 1 nm bandwidth and 1 nm step size. Four scans were taken per measurement and were averaged and smoothed using Savitsky–Golay smoothing with a window size of three. A background spectrum of the CD buffer was obtained and subtracted from the peptide spectra. CD values were converted to mean residue ellipticity [θ] (deg cm$^2$ dmol$^{-1}$) from mdeg, and the K2D3 method was used to determine % helicities[77].

## *Xenopus* egg extract licensing assays

Whole metaphase-arrested *Xenopus* egg extract and demembranated sperm nuclei were prepared as described[78]. Female frogs were primed with 50 units of Folligon (Pregnant Mare Serum Gonadotrophin) 3 days prior to the eggs being required to increase the number of stage 6 mature oocytes and 2 days later, were injected with 500 units Chorulon (Chorionic Gonadotrophin, Intervet) to induce ovulation. Twice-injected frogs were placed in individual laying tanks at 18–21 °C in 2 L 1× MMR egg laying buffer, prepared from a 10× stock (1 M NaCl, 20 mM KCl, 10 mM MgCl$_2$, 20 mM CaCl$_2$, 1 mM EDTA, 50 mM HEPES-NaOH, pH 7.8). The following morning, eggs were collected, graded and usable eggs rinsed in 1× MMR to remove any non-egg debris. Washed eggs

were dejellied in 2% (w/v) cysteine (pH 7.8), washed in XBE2 (1× XB salts, 1.71% (w/v) sucrose, 5 mM K-EGTA, 10 mM HEPES-KOH, pH 7.7; 10× XB salts: 2 M KCl, 40 mM MgCl$_2$, 2 mM CaCl$_2$) and then into XBE2 containing 10 μg/ml leupeptin, pepstatin and aprotinin. Dejellied and washed eggs were centrifuged in 14 ml tubes, containing 1 ml XBE2 plus protease inhibitors containing 100 μg/ml cytochalasin D, at 1400 x $g$ in a swinging bucket rotor for 1 min at 16 °C to pack the eggs, after which excess buffer and dead eggs which rise to the surface were removed from the top of each tube. Packed eggs were crushed by centrifugation at 16,000 x $g$ in a swinging bucket rotor for 10 min at 16 °C. The dirty brown cytoplasmic layer was collected using a 20 G needle and a 1 ml syringe via side puncture. From this point onwards the extract was kept on ice. The crude extract was supplemented with cytochalasin D, leupeptin, pepstatin and aprotinin, all to a final concentration of 10 μg/ml, The extract was clarified by centrifugation at 84,000 x $g$ in a pre-cooled SW55 rotor swinging bucket rotor at 4 °C for 20 min. The golden cytoplasmic layer was recovered, supplemented with glycerol to 2% (v/v) and frozen in aliquots in liquid nitrogen and stored at −80 °C until required.

Sperm was recovered from testes isolated from male frogs postmortem following a lethal dose of anaesthetic (0.2% (w/v) Tricaine mesylate MS222, -0.5% (w/v) NaHCO$_3$ to pH 7.5). Isolated testes were washed carefully to avoid bursting in EB (50 mM KCl, 5 mM MgCl$_2$, 2 mM dithiothreitol or β-mercaptoethanol, 50 mM HEPES-KOH, pH 7.6), prior to being finely chopped with a clean razor blade in fresh EB. Recovered lysate was filtered through a 25 μm nylon membrane to remove particulate matter. Filtered sperm was centrifuged at 2000 x $g$ at 4 °C for 5 min; selective resuspension of the sperm pellet allowed separation of the sperm from contaminating erythrocytes; the resuspended sperm was re-spun and the pellet resuspended in 0.5 ml SuNaSp (0.25 M sucrose, 75 mM NaCl, 0.5 mM spermidine, 0.15 mM spermine, 15 mM HEPES-KOH, pH 7.6) per testis. The sperm was demembranated with the addition of 25 μl per testis lysolecithin (5 mg/ml, in H$_2$O) for 10 min at RT. Demembranated sperm were respun and resuspended in SuNaSp plus 3% (w/v) BSA to quench the demembranation reaction. Quenched sperm were respun and resuspended in 100 μl EB plus 30% glycerol per testis, counted using a haemocytometer and stored at −80 °C.

The *Xenopus* egg extract replication licensing assay is performed in 2 stages: stage 1, the 'replication licensing stage' in which replication licensing but not DNA replication is permissible and the second stage, the 'DNA replication stage', in which DNA replication is supported but replication licensing is inhibited by the addition of 15 nM non-destructable geminin$^{DEL}$ to the extract. The Replication Licensing Stage: 80 ng (0.5 μl) of demembranated *Xenopus* sperm nuclei (160 ng/μl) was incubated for 20 min at 20 °C in 2.5 μl of a five-fold diluted membrane-free interphase extract (see below) to support replication licensing, ± the geminin constructs to be assayed. The replication stage: since a diluted extract cannot support nuclear assembly and therefore DNA replication initiation, after the 20 min licensing period 6 μl of whole interphase egg extract, supplemented with 15 nM geminin$^{DEL}$ to inhibit replication licensing in the second stage of the reaction, is added to each licensing reaction to support replication initiation at licensed sites on the DNA and the reaction incubated for 60 min at 20 °C.

The extent of DNA synthesis was assayed by measuring the incorporation of [α–$^{32}$P] dATP into acid-insoluble material followed by scintillation counting. 50 nCi/μl [α–$^{32}$P] dATP from a high activity (10 mCi/ml) stock was added to the geminin$^{DEL}$-supplemented whole interphase egg extract to be used in the second 'DNA replication stage' of the replication licensing assay. Replication reactions were stopped at 60 min by the addition of 160 μl Stop-C (0.5% (w/v) SDS, 5 mM EGTA, 20 mM Tris HCl, pH 7.5) plus freshly added 0.2 mg ml Proteinase K (from stock of 20 mg/ml proteinase K, 50% (v/v) glycerol, 10 mM Tris–HCl, pH 7.5) and incubated at 37 °C for 30 min. Samples were precipitated at 4 °C for 30 min by the addition of 4 ml 10% TCA (10%

(w/v) TCA, 2% (w/v) $Na_4P_2O_7 \cdot 10H_2O$). Forty microlitres (1% of 4 ml) of the total reaction were spotted on a paper disc and acid-insoluble material was recovered from solution by filtration through a glass fibre filter mounted on a vacuum manifold. The glass fibre filters were twice washed in 5% TCA (5% (w/v) TCA, 0.5% (w/v) $Na_4P_2O_7.10H_2O$), once in 100% ethanol, and then air dried. The paper and glass fibre filters were then quantified by scintillation counting. Precipitated material was expressed as a percentage of total counts (%TC) from which DNA replication (ng/µl) was calculated by multiplying by a factor of 0.654.

A positive control (80 ng sperm nuclei mock-licensed in LFB 1/50 + 2.5 mM ATP for 20 min during the 'replication licensing' stage to which untreated whole interphase egg extract was added for stage 2) and a negative control (80 ng sperm nuclei mock-licensed in LFB 1/50 + 2.5 mM ATP for 20 min prior to the addition of geminin[DEL]-treated interphase extract) were included in each reaction. The kinetics of the extracts to be used in the assay were determined so that the replication reaction was 'stopped' when 90% of the DNA was replicated in control reactions such that no 'catch up' DNA replication in any inhibited samples could occur, thus avoiding skewing of the results. *Xenopus* egg extract licensing assays were performed using peptide concentrations between 15 µM and 15 nM.

Metaphase-arrested egg extract was supplemented with 250 µg/ml cycloheximide, 25 mM phosphocreatine and 15 µg/ml creatine phosphokinase and incubated with 0.3 mM $CaCl_2$ for 15 min to trigger release from metaphase arrest and promote entry into interphase. The membrane-free extract used in stage 1 of the replication licensing assay was prepared by the five-fold dilution in LFB1/50 supplemented with 2.5 mM Mg-ATP of the above calcium-released whole interphase egg extract. The diluted interphase extract was centrifuged at 186,000 x g, at 4 °C for 20 min in a TLA-100 rotor; the supernatant was collected, aliquoted, frozen in liquid nitrogen and stored at −70 °C until required. All assays presented were performed with the same diluted extract. For the second stage of the licensing assay the above calcium-released interphase extract was supplemented with 15 nM geminin[DEL] and 50 nCi/µl [α–$^{32}$P] dATP and used immediately.

### Reporting summary

Further information on research design is available in the Nature Portfolio Reporting Summary linked to this article.

## Data availability

Data are available within the article and the Supplementary Information files. Data and materials can be obtained from the corresponding authors upon request. Source data are provided with this paper.

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

## Acknowledgements

The electron microscopy data was collected at the EM facility of the MRC LMS Institute, London, with the support of Ricardo Aramayo. We would like to thank the Speck lab, Yasunori Noguchi (Kyushu University) and Luitpold Maximilian Reuter (IMB Mainz) for critical reading of the manuscript. Thank you to Ian Brennan for help with Python scripts used to analyse AlphaFold output data. Thank you to Professor Dek Woolfson and Dr Bram Mylemans (University of Bristol) for useful discussions about the Geminin coiled coil. The work was supported by Cancer Research UK (DRCNPG-May21\100006), awarded to C.S. A.B. was supported by a Sir Henry Dale Fellowship jointly funded by the Wellcome Trust and the Royal Society (213425/Z/18/Z). J.T. acknowledges funding from a Department of Chemistry PhD Scholarship from Imperial College London. A.R.B. was supported by a Cancer Research UK Career Development Fellowship (C63833/A25729). R.C. was supported by a FUNDAME Postdoctoral Fellowship and core-funding from the MRC LMS to the Cell Cycle Control team (MC-A658-5TY60).

## Author contributions

A.R.B., A.B. and C.S. supervised and conceived the project. J.T., L.E., V.L., S.F. and P.G. designed the experimental approaches. J.T., L.E., V.L., M.P., H.B., N.S., R.C. and S.F. performed the biochemical assays. J.T., L.E. and S.F. carried out statistical analysis of biochemical assays. M.P. carried out Mass Photometry experiments. A.S. performed electron microscopy and data analysis. P.G. performed *Xenopus* egg extract experiments. J.T., L.E., S.F. and C.S. interpreted the biochemical data. J.T., L.E., P.G., M.P. and S.F. prepared figures. J.T., L.E., S.F. and C.S. wrote the manuscript with critical input from all authors. J.B., A.R.B., A.B. and C.S. acquired funding for this study.

## Competing interests

The authors declare no competing interests.
