## [Transparent Peer Review file · Nature Communications]

Geminin inhibits DNA replication licensing by sterically blocking CDT1-MCM2 interactions

Corresponding Author: Professor Christian Speck

Version 0:

Reviewer comments:

Reviewer #1

(Remarks to the Author)

The manuscript by Speck and colleagues describes studies addressing the mechanism of geminin function, a key regulator of metazoan DNA replication. It has been known for some time that Geminin acts to inhibit origin licensing by binding Cdt1. It has generally been thought that this binding event interfered with the Cdt1-Mcm2-7 binding interface preventing its participation in the licensing event. Here the authors use a combination of AlphaFold predictions, a biochemical assay for human origin licensing, and mutagenesis to demonstrate that Geminin binding prevents Cdt1 binding to Mcm2-7 through less direct mechanism. Along the way, the authors also identify additional structural elements important for geminin-Cdt1 interactions and the inhibition of origin licensing.

The first part of the paper describes studies addressing the role of several different elements of the Geminin-Cdt1 structure in origin licensing. The authors nicely address the role of each of these domains showing that some (regions 1 and 2) are not important for geminin function and that others (regions 3.1 and 3.2) are. In each case, they use a combination of geminin-Cdt1 interaction studies and reconstituted origin licensing reactions to assess the importance of the region. These are well-executed convincing studies.

The authors go on to demonstrate that geminin inhibits the origin licensing process prior to the formation of the human OCCM origin licensing intermediate. Using docking of their predicted geminin-Cdt1 structure to the OCCM they develop the hypothesis that geminin does not directly inhibit the Cdt1-Mcm2-7 interaction. Instead, they propose and nicely test a model in which a region of geminin not involved in binding Cdt1 sterically clashes with the Mcm2 subunit to prevent binding. In particular, they show that mutants that eliminate the clashing region of geminin retain their ability to interact with Cdt1 at high affinity but show greatly reduced ability to inhibit origin licensing. These data are both surprising and convincing.

The last part of the paper demonstrates that CDK also contributes to the inhibition of origin licensing in the reconstituted assay. Although these studies show that (in vitro) CDK can inhibit origin licensing, they are less convincing than the remainder of the studies as it is not clear what the CDK target is and whether that target is relevant to in vivo CDK regulation of origin licensing. It is also noteworthy that there is no in vivo data to support the authors' proposed mechanism of geminin function. While it is not essential that this be included in the current manuscript, it would be nice for the authors to discuss whether there are mutations in the literature that would support their findings (e.g. Meier Gorlin mutants).

Overall, this is a well executed study that provides new insights into the regulation of an essential event in DNA replication. The results identify critical new information about the geminin-Cdt1 interaction and how that inhibits origin licensing and will be of interest to those studying DNA replication and cell cycle regulation.

Specific point:

The section discussing region 3.1 should be clarified. The authors refer to the targeted region in multiple ways and (coiled-coil, C-terminal domain) and fail to remind the reader that they are talking about geminin and not Cdt1 until late in the paragraph. This is important because both parts of Cdt1 and geminin are included in the region 3 box.

Reviewer #2

(Remarks to the Author)

This study investigates the inhibitory role of Geminin during the replication licensing process. Utilizing their established system with purified human proteins, the authors address this important question through a biochemical approach. They examine how Geminin affects various steps in replication licensing, including the recruitment and loading of MCM2-7, and seek to characterize the oligomeric state and structural domains of the Cdt1–Geminin complex required for inhibition. While the central claim of the paper is supported by the data, there are notable conceptual and experimental limitations that constrain the study's potential impact and contribution to the field.

Conceptually, certain results from the licensing assay are presented in a way that may be confusing or unclear, potentially making it difficult for readers to fully follow the narrative. Technically, while some of the most intriguing and novel findings are under explored, the manuscript places disproportionate emphasis on negative or less relevant results. Together, these issues somewhat obscure the clarity and diminish the perceived novelty of the study's overall message.

Major Conceptual Points

1. Conceptual definition and interpretation of preRC assay outcomes.

The manuscript investigates three licensing outcomes under different experimental conditions:

- High-salt wash with ATP (HS or typically referred to as "loading").
- Low-salt wash with ATP (LS or typically referred ATP LSW or "release").
- Low-salt wash with ATP γ S (typically referred to as "recruitment").

While the authors correctly interpret the loading and recruitment conditions (Figures 1 and 7, respectively), the treatment and interpretation of the low-salt ATP condition (Figure 1) are confusing and conceptually problematic. This condition is known to retain mainly DHs, ORC and others intermediates (e.g., MO, OM complexes). Referring to this condition as "recruitment" is inappropriate and reflects a lack of precision. Due to its complex nature, the interpretation of this process requires more careful consideration. For example, this misinterpretation leads to internal inconsistencies—for instance, the authors claim that MCM2-7 can be recruited without Cdt1, yet they proceed to analyze Cdt1 domains involved in MCM2-7 recruitment. This contradiction also undermines the relevance of Figure 7, since Geminin interacts exclusively with Cdt1.

2- Inaccurate and oversimplified Licensing Model

The model in Figure 1a is conceptually inaccurate. The OCCM complex is only observed when a slowly hydrolysable ATP analogue (ATP γ S) is used, as ATP hydrolysis triggers MCM ring closure and subsequent loading. Consequently, the OCCM is not equivalent to the first MCM loading step, as the figure suggests. Furthermore, it is not expected to be present under ATP conditions, contrary to what is proposed in the text and figure.

Additionally, the authors imply that DH formation proceeds through a single, MO-dependent pathway. However, prior work—including from Speck's group—demonstrates that DHs can assemble via MO-independent routes. This is further supported by the recent *in vitro* results that have shown that O6 is not essential for DH assembly. These findings should be integrated to avoid an overly restrictive interpretation of the licensing mechanism.

Major technical Points

1. Detection and Visualization of Cdt1

There is a discrepancy in Cdt1 detectability between Figures 1 and 7. In Figure 1, under low-salt ATP conditions, Cdt1 is barely visible without signal amplification by western. However, in Figure 7, under ATP γ S conditions, the Cdt1 band is clearly detected despite co-migrating with two ORC subunits (Figure 7a, lanes 1 and 2). In addition, the lanes 1 in 7a and 7c should be identical, yet Cdt1 is only present in 7a. This inconsistency should be addressed and the same detection method should be used in both figures to make them comparable.

2. Excess Cdc6 Usage

The authors use a substantial excess of Cdc6 in their assays (evident in Figures 1c and 1j, input lanes). If stoichiometric amounts are non-functional, it raises questions about the activity or stability of the purified protein. Consequently, conclusions regarding Cdc6 stabilization by full-length geminin should be interpreted with caution and more thoroughly validated.

3. Oligomeric State of the Cdt1–Geminin Complex

The data indicate that excess geminin, x2 or more, inhibits double hexamer (DH) formation, independent of the degree of excess. However, the mass photometry results suggest that only specific stoichiometries, namely 1:2 and 2:4 (Cdt1: geminin), support the formation of heterotrimeric complexes. It is unexpected that higher geminin ratios, such as 1:3 and 1:4, do not yield detectable heterotrimers, if this structure is the cause of inhibition.

To clarify the functional relevance of these observations, it would be important to test the 1:3 and 1:4 Cdt1: geminin ratios in

the licensing assay. This would help determine whether formation of the heterotrimeric complex is necessary for inhibition or not. Conversely, analysing higher geminin excess conditions (e.g., 6-fold and 18-fold) by mass photometry would further confirm the requirement for heterotrimeric complex formation.

4. Limited functional relevance of AlphaFold-predicted interactions to geminin activity

The interactions between Cdt1 and geminin predicted by AlphaFold do not appear to reflect novel functional contacts. When tested in the licensing assay, these predicted interfaces did not exhibit a significant role in geminin function. Nevertheless, the manuscript devotes three (Fig. 3-5) of the nine main figures to these predictions, which seems disproportionate given their limited functional contribution.

In contrast, Figure 6 presents more compelling data, where the authors generate a series of geminin truncations to define the minimal region necessary for activity in both the human licensing assay and *Xenopus* egg extracts. These experiments convincingly demonstrate that truncation beyond threonine 82 eliminates geminin function—consistent with existing structural data, as T82 corresponds to the N-terminal residue resolved in the Cdt1–geminin crystal structure (Lee et al., 2004).

To strengthen this conclusion, it would be valuable to test whether mutating the corresponding interaction surface on Cdt1 similarly affects function, thereby providing complementary support from the Cdt1 side.

5. CDK/Cyclin-Mediated Inhibition of Licensing

CDK/cyclin complexes phosphorylate multiple components of the preRC, not just Cdt1. Therefore, the inhibition observed in Figure 9 may result from CDK-mediated phosphorylation of several preRC factors, not solely Cdt1. The authors should specifically address whether CDK-dependent phosphorylation of Cdt1 alone is sufficient to inhibit licensing in the presence of geminin. Clarifying the contribution of Cdt1 phosphorylation relative to other CDK targets would improve the mechanistic interpretation of the presented model.

6. Confusing Nomenclature for Cdt1 Constructs

The designation of the truncated version simply as “Cdt1” is both careless and misleading. According to standard conventions, the unmodified (wild-type) protein should be labeled as “Cdt1” or “FL” when used in conjunction with truncated constructs, while truncations should be named based on the specific deleted domain. It is important to clarify this nomenclature to prevent confusion among readers. For instance, the manuscript refers to the “N-terminus” of the truncated “Cdt1” construct, although this region actually corresponds to the middle helical domain (MHD) of the wild-type or full-length Cdt1 protein.

Minor Points

- In the presentation of the preRC assay results, the names of the individual proteins should be clearly indicated.
- Remove duplicated conclusions. One paragraph states: "As such, we define the minimal geminin coiled-coil section that is required for CDT1 binding as residues 96–135." The following paragraph repeats: "Subsequently, we define Gem96–135 as the minimal region of geminin that is capable of binding to CDT1." These should be consolidated or rephrased to avoid redundancy.
- Use the term coiled-coil rather than coil-coil throughout the manuscript.
- In Figure 1j, the input lanes should also show the 5× loading condition for comparison.
- In Figure 8a, use distinct colours to clearly highlight the geminin residues that sterically clash with Mcm2.

(Remarks to the Author)

This study by Tomkins and colleagues follows up on recent work by a subset of these authors to reconstitute human origin licensing *in vitro*. Having established that assay in their 2024 paper, they now explore the mechanism by which human geminin blocks that reaction. They identify the precise step in preRC assembly suppressed by geminin that requires ATP hydrolysis and allows CDT1 binding, CDC6 release, and ORC6 recruitment during the first MCM loading. They provide evidence to support the stoichiometry of the CDT1-geminin complex (1:2) as opposed to higher-order complexes which had been suggested earlier. There were reports by several others that the mechanism is blocking CDT1-MCM binding (for example, PubMed 12192004), so this model is not completely unanticipated. The claim that “the mechanism of this inhibition has remained unknown for the last 25 years” is perhaps a bit too strong. Nonetheless, these experiments show that the mechanism is not simply geminin obscuring the MCM interaction sites on CDT1.

The authors then conduct a series of experiments with geminin peptides to define the regions required for inhibition and that that simply binding CDT1 is not enough to inhibiting MCM loading. The results in Figure 7 showing that geminin can rapidly remove CDT1 from an existing intermediate complex are novel because one would have thought that once MCM is recruited, geminin would have minimal effect; this fits with the authors' previous result that CDT1 is not required for the first MCM to be recruited. The experimental data are impressively clean, and the inclusion of multiple replicates adds rigor.

The authors used synthetic geminin peptides of varying lengths to dissect interactions and their consequences for *in vitro* MCM loading. One peptide, 76-135, binds CDT1 almost as well as full-length geminin, but does not effectively block MCM loading. Incrementally longer peptides are increasingly better at inhibiting MCM loading without necessarily binding CDT1 better. This separation of CDT1 binding from MCM loading inhibition is an interesting contribution to understanding human licensing. The authors propose a model, inspired by AlphaFold3, whereby the geometry and rigidity of the CDT1-geminin complex creates a steric clash between the c-terminal region of the geminin coiled-coil and the MCM2 subunit.

Finally, the results in Figure 9 that show geminin alone is not enough to block MCM loading, but that CDK-mediated phosphorylation of CDT1 is required in collaboration is satisfying within the context of current models of MCM loading control. Overall, this study is a nice addition to our understanding an important aspect of genome stability since geminin is a major player in preventing re-replication. It is also an example of beautiful biochemistry with a speculative model. There are some aspects that the authors should address:

1. There is insufficient rationale or interpretation of the experiments in Figure 1 i-l that mix the individual domains of CDT1 – what do the authors think is happening in those reactions, and is the effect meaningful? Does it suggest anything about what human CDT1 does in the human preRC reactions?

2. It isn't clear what the importance of the CDT1 “region 1” helix predicted by AlphaFold3 is since there is no detectable difference between full-length CDT1 and a version lacking the entire N-terminus. The authors devote 2 main figures to this predicted interaction, but do not show that it occurs or is consequential (*in vitro*). Alternatively, the authors should state if they think this predicted interaction is real, and if so, what it might be for.

3. Another separation of CDT1 binding from geminin function allele that changes two amino acids was described by Suchyta et al (PMID: 25988259). How does this allele fit with the authors' model?

4. One experiment to try to directly test the clash model predicted by AlphaFold3 is in supplemental Figure 17. Although creative, I'm not convinced this glycine insertion demonstrates that the angle of geminin relative to MCM is the critical molecular feature. The argument is that CDT1 “positions the geminin coiled-coil at the correct angle to generate the geminin-MCM2 clash.” The LINK mutant binds CDT1 about 3 times less well compared to wild-type, and there's no experiments that test the angle or that CDT1 itself uses the predicted binding site to create that angle. The model remains speculative and should be presented as such.

5. The dissection of the geminin coiled-coil region is only through changing the length and not the sequence. There is a strong, yet unacknowledged, assumption that the specific amino acids are irrelevant and only the length matters. If the interpretation is correct, then any sequence that forms a coiled-coil could substitute. The authors should address this point in some way.

Minor:

a) Is the length of the geminin coiled-coil region that is predicted to clash with MCM2 conserved?

b) A brief discussion of the limitations of the study would be helpful for future readers. For example CDT1 and geminin were produced in bacteria, but both proteins are reported to be decorated with post-translational modifications in cells (PhosphoSitePlus), the loading reactions use purified DNA, etc.

Version 1:

Reviewer comments:

Reviewer #1

(Remarks to the Author)

The revised manuscript from Speck and colleagues has done an excellent job of responding to my concerns. The addition of the experiments examining the effects of CDK phosphorylation of individual origin licensing proteins greatly improves the last part of the paper. The changes in the text in response to the reviews has also greatly clarified the manuscript. This work will be an excellent addition to the literature and of interest to those studying the DNA replication, the cell cycle and the molecular mechanisms of complex assembly events in the cell.

Reviewer #2

(Remarks to the Author)

Despite addressing most of the points previously raised by this reviewer in a positive manner, I still have significant concerns regarding the superficial treatment of some of the novel findings, as well as the disproportionate emphasis placed on results that are less relevant to the main conclusions of the study, as also noted by Reviewer 3.

Regarding the lack of controls for novel findings. The question of how CDK regulates licensing in human cells remains one of the most critical and incompletely understood issues in the field, as it is essential for proper regulation of chromosome replication. Therefore, if the authors choose to address this topic, it must be done with appropriate rigor, including, at a minimum, phosphorylation controls. In Figure 8, the authors present results from pre-incubating pre-RC components with CDK; however, they provide no evidence confirming that these proteins were actually phosphorylated. Moreover, they do not assess the effectiveness of the CDK inhibitor, which is essential to ensure that the CDK added later in the reaction is fully inhibited. These missing controls are critical for meaningful interpretation of the results and to avoid further misunderstanding in the field.

Concerning the disproportionate emphasis on less relevant results. The geminin deletions and/or Cdt1 point mutants described in Figure 5, the only structural or AlphaFold-based figure showing positive results, could have been designed in exactly the same way using the crystal structure published by Lee et al., Nature 2004, which clearly identifies all key residues involved in the interaction. Therefore, the sentence in the abstract—"AlphaFold modelling revealed a critical N-terminal CDT1-binding helix and loop in geminin, which proved essential for inhibition"—while appealing, should be removed. If the authors choose to retain it, they must clearly state that these findings are consistent with the previously published structural data by Lee et al., to avoid overstating the novelty of their results.

Reviewer #3

(Remarks to the Author)

The revised version is improved and satisfies our requests. The authors should be congratulated.

Version 2:

Reviewer comments:

Reviewer #2

(Remarks to the Author)

The authors have satisfactorily addressed all of my previous concerns, and the revised manuscript shows significant improvement in clarity and scientific rigor.

We want to thank all the reviewers for their time spent reviewing the manuscript and providing constructive feedback. This has greatly helped us to improve the manuscript.

Reviewer #1

The manuscript by Speck and colleagues describes studies addressing the mechanism of geminin function, a key regulator of metazoan DNA replication. It has been known for some time that Geminin acts to inhibit origin licensing by binding Cdt1. It has generally been thought that this binding event interfered with the Cdt1-Mcm2-7 binding interface preventing its participation in the licensing event. Here the authors use a combination of AlphaFold predictions, a biochemical assay for human origin licensing, and mutagenesis to demonstrate that Geminin binding prevents Cdt1 binding to Mcm2-7 through less direct mechanism. Along the way, the authors also identify additional structural elements important for geminin-Cdt1 interactions and the inhibition of origin licensing.

The first part of the paper describes studies addressing the role of several different elements of the Geminin-Cdt1 structure in origin licensing. The authors nicely address the role of each of these domains showing that some (regions 1 and 2) are not important for geminin function and that others (regions 3.1 and 3.2) are. In each case, they use a combination of geminin-Cdt1 interaction studies and reconstituted origin licensing reactions to assess the importance of the region. These are well-executed convincing studies.

The authors go on to demonstrate that geminin inhibits the origin licensing process prior to the formation of the human OCCM origin licensing intermediate. Using docking of their predicted geminin-Cdt1 structure to the OCCM they develop the hypothesis that geminin does not directly inhibit the Cdt1-Mcm2-7 interaction. Instead, they propose and nicely test a model in which a region of geminin not involved in binding Cdt1 sterically clashes with the Mcm2 subunit to prevent binding. In particular, they show that mutants that eliminate the clashing region of geminin retain their ability to interact with Cdt1 at high affinity but show greatly reduced ability to inhibit origin licensing. These data are both surprising and convincing.

The last part of the paper demonstrates that CDK also contributes to the inhibition of origin licensing in the reconstituted assay. Although these studies show that (in vitro) CDK can inhibit origin licensing, they are less convincing than the remainder of the studies as it is not clear what the CDK target is and whether that target is relevant to in vivo CDK regulation of origin licensing. It is also noteworthy that there is no in vivo data to support the authors' proposed mechanism of geminin function. While it is not essential that this be included in the current manuscript, it

would be nice for the authors to discuss whether there are mutations in the literature that would support their findings (e.g. Meier Gorlin mutants).

Overall, this is a well executed study that provides new insights into the regulation of an essential event in DNA replication. The results identify critical new information about the geminin-Cdt1 interaction and how that inhibits origin licensing and will be of interest to those studying DNA replication and cell cycle regulation.

We thank the reviewer for their positive comments about our manuscript. We have now included an experiment where individual pre-RC proteins have been CDK-phosphorylated (Figure 8). These data indicate that in the context of our assay, CDK mainly targets ORC1-5 and CDC6. Moreover, we show that CDK-mediated regulation of ORC1-5 and CDC6 in combination with geminin prevents helicase loading. We have updated the text to include these findings. Considering that ORC1 and CDC6 are known targets of CDK regulation in both the human and yeast systems, and it is well established that CDKs inhibit DNA licensing in yeast via ORC and Cdc6, this result makes sense (Hossain et al, Molecular Cell 2021; Hossain and Stillman, G&D 2012; Nguyen et al, Nature 2001). Also, we discuss a specific publication focused on ORC1 Meier Gorlin mutations that affect the interaction with cyclin E-CDK2, as this study hints that CDK regulation affects DNA licensing in humans (Hossain and Stillman, G&D, 2012). Moreover, we mention that Cyclin E and A overexpression is associated with misregulation of DNA licensing and cancer (Ekholm-Reed et al, JCB 2004; Tane and Chibazakura, Cell cycle 2009).

Specific point:

The section discussing region 3.1 should be clarified. The authors refer to the targeted region in multiple ways and (coiled-coil, C-terminal domain) and fail to remind the reader that they are talking about geminin and not Cdt1 until late in the paragraph. This is important because both parts of Cdt1 and geminin are included in the region 3 box.

Thank you for the comment. We have clarified in the text that the geminin coiled-coil is the region of interest, modified the figure panel to remove as much of CDT1 from the box as possible and changed the figure legend of Figure 4a.

Reviewer #2 (Remarks to the Author):

This study investigates the inhibitory role of Geminin during the replication licensing process. Utilizing their established system with purified human proteins, the authors address this important question through a biochemical approach. They examine how Geminin affects various steps in replication licensing, including the recruitment and loading of MCM2-7, and seek to characterize the oligomeric state and structural domains of the Cdt1-Geminin complex required for inhibition. While the central claim of the paper is supported by the data, there are notable conceptual and experimental limitations that constrain the study's potential impact and contribution to the field.

Conceptually, certain results from the licensing assay are presented in a way that may be confusing or unclear, potentially making it difficult for readers to fully follow the narrative. Technically, while some of the most intriguing and novel findings are under explored, the manuscript places disproportionate emphasis on negative or less relevant results. Together, these issues somewhat obscure the clarity and diminish the perceived novelty of the study's overall message.

We thank the reviewer for their thorough and kind response and have addressed the points they raised.

Major Conceptual Points

1. Conceptual definition and interpretation of preRC assay outcomes.

The manuscript investigates three licensing outcomes under different experimental conditions:

- High-salt wash with ATP (HS or typically referred to as “loading”).
- Low-salt wash with ATP (LS or typically referred ATP LSW or “release”).
- Low-salt wash with ATP γ S (typically referred to as “recruitment”).

While the authors correctly interpret the loading and recruitment conditions (Figures 1 and 6, respectively), the treatment and interpretation of the low-salt ATP condition (Figure 1) are confusing and conceptually problematic. This condition is known to retain mainly DHs, ORC and others intermediates (e.g., MO, OM complexes). Referring to this condition as “recruitment” is inappropriate and reflects a lack of precision. Due to its complex nature, the interpretation of this process requires more careful consideration. For example, this misinterpretation leads to internal inconsistencies—for instance, the authors claim that MCM2-7 can be recruited without

Cdt1, yet they proceed to analyze Cdt1 domains involved in MCM2-7 recruitment. This contradiction also undermines the relevance of Figure 7, since Geminin interacts exclusively with Cdt1.

We thank the reviewer for their comments and suggestions on how to improve our manuscript. We removed the term recruitment for the LS wash conditions in the text and Figures 1 and 6, as it is confusing.

2- Inaccurate and oversimplified Licensing Model

The model in Figure 1a is conceptually inaccurate. The OCCM complex is only observed when a slowly hydrolysable ATP analogue (ATP γ S) is used, as ATP hydrolysis triggers MCM ring closure and subsequent loading. Consequently, the OCCM is not equivalent to the first MCM loading step, as the figure suggests. Furthermore, it is not expected to be present under ATP conditions, contrary to what is proposed in the text and figure. Additionally, the authors imply that DH formation proceeds through a single, MO-dependent pathway. However, prior work—including from Speck's group—demonstrates that DHs can assemble via MO-independent routes. This is further supported by the recent *in vitro* results that have shown that O6 is not essential for DH assembly. These findings should be integrated to avoid an overly restrictive interpretation of the licensing mechanism.

We appreciate the reviewer's concern over the diagram. We revised Figure 1a and labelled the ATP γ S section as enriched for OCCM. In the text, we describe the MCM2-7 hexamer as part of the OCCM as topologically encircled by DNA.

Based on extensive literature, the OCCM complex is an intermediate in the ATP-hydrolysis-dependent multi-step helicase loading pathway (reviewed in Costa and Diffley, Annual Reviews, 2022). It also serves as an essential reference point to understand at which step geminin inhibits the reaction; therefore, we included it in the figure.

As suggested, we have added the ORC6-dependent pathway to Figure 1a and added a full schematic of human DNA licensing as Supplementary Figure 1.

Major technical Points

1. Detection and Visualization of Cdt1

There is a discrepancy in Cdt1 detectability between Figures 1 and 6. In Figure 1, under low-salt ATP conditions, Cdt1 is barely visible without signal amplification by western. However, in Figure 6, under ATP γ S conditions, the Cdt1 band is clearly detected despite co-migrating with two ORC

subunits (Figure 6a, lanes 1 and 2). In addition, the lanes 1 in 6a and 6c should be identical, yet Cdt1 is only present in 6a. This inconsistency should be addressed and the same detection method should be used in both figures to make them comparable.

We thank the reviewer for their detailed analysis. In Figure 6, the reaction has been performed in the presence of ATPyS, which prevents the release of CDT1 (Wells et al, Nature Communications 2025). Therefore, the CDT1 band is much more prominent.

We also appreciate the comment regarding Figures 6a and 6c. In one experiment, we used an old batch of ATPyS; hence, we repeated the experiment. The new experiment with fresh ATPyS shows identical CDT1 levels in Figures 6a and 6c. Historically, we used western blotting before fluorescence for CDT1 detection. As both approaches yield identical results, we refrain from making any changes.

2. Excess Cdc6 Usage

The authors use a substantial excess of Cdc6 in their assays (evident in Figures 1c and 1j, input lanes). If stoichiometric amounts are non-functional, it raises questions about the activity or stability of the purified protein. Consequently, conclusions regarding Cdc6 stabilization by full-length geminin should be interpreted with caution and more thoroughly validated.

We acknowledge the reviewer's concerns. In budding yeast, optimal ORC-Cdc6 complex formation is only observed when a two-fold excess of Cdc6 is used (Speck et al NSMB 2006). We have therefore carried this aspect of the protocol over to the human pre-RC (Wells et al Nature Communications 2025). That said, we performed a pre-RC assay with half the amount of CDC6 (47 nM instead of 94 nM) to assess the protein's activity. With stoichiometric amounts of ORC1-5 and CDC6, we still observe a stabilisation of CDC6 and a reduction of ORC6 in the presence of Geminin. This is now shown in Supplementary Figure 3a.

3. Oligomeric State of the Cdt1–Geminin Complex

The data indicate that excess geminin, x2 or more, inhibits double hexamer (DH) formation, independent of the degree of excess. However, the mass photometry results suggest that only specific stoichiometries, namely 1:2 and 2:4 (Cdt1: geminin), support the formation of heterotrimeric complexes. It is unexpected that higher geminin ratios, such as 1:3 and 1:4, do not yield detectable heterotrimers, if this structure is the cause of inhibition. To clarify the functional relevance of these observations, it would be important to test the 1:3 and 1:4 Cdt1: geminin ratios in the licensing assay. This would help determine whether formation

of the heterotrimeric complex is necessary for inhibition or not. Conversely, analysing higher geminin excess conditions (e.g., 6-fold and 18-fold) by mass photometry would further confirm the requirement for heterotrimeric complex formation.

Thank you for your comment. In these mass photometry experiments, the excess of geminin leads to proportionally more counts of uncomplexed molecules. Mass-photometry is not ideally suited to study asymmetric ratios of proteins, as the method has no enrichment for specific molecular weights; thus it is different to gel-filtration. Therefore, the impression arises that less complex is formed. We have added a clarification in the manuscript. Also, please consider that Figures 2b and c have an increasing excess of geminin, but only a slight increase in inhibition is seen after 2x.

Nevertheless, we attempted to increase the geminin concentrations to 6- and 18-fold in mass-photometry experiments, but as expected, this leads to a massive peak of uncomplexed geminin, and only a very minor peak for the complex which hinders the interpretation of the data. For these reasons, we did not include the data.

4. Limited functional relevance of AlphaFold-predicted interactions to geminin activity

The interactions between Cdt1 and geminin predicted by AlphaFold do not appear to reflect novel functional contacts. When tested in the licensing assay, these predicted interfaces did not exhibit a significant role in geminin function. Nevertheless, the manuscript devotes three (Fig. 3-5) of the nine main figures to these predictions, which seems disproportionate given their limited functional contribution.

We thank the reviewer for the comment. Our study benefits from a systematic approach, providing a complete analysis that will serve as a reference for the field. That said, we understand the reviewer's concern and accordingly consolidated Figures 3 and 4 to devote less space to the negative AlphaFold results. Whilst region 3.1 (Figure 4) may be classed as a negative result, it lays the foundation for the minimal interaction region between CDT1 and geminin.

In contrast, Figure 5 presents more compelling data, where the authors generate a series of geminin truncations to define the minimal region necessary for activity in both the human licensing assay and *Xenopus* egg extracts. These experiments convincingly demonstrate that truncation beyond threonine 82 eliminates geminin function—consistent with existing structural

data, as T82 corresponds to the N-terminal residue resolved in the Cdt1–geminin crystal structure (Lee et al., 2004).

To strengthen this conclusion, it would be valuable to test whether mutating the corresponding interaction surface on Cdt1 similarly affects function, thereby providing complementary support from the Cdt1 side.

We thank the reviewer for this suggestion. The BUDE alanine scan results (Supplementary Figure 14) were used to identify critical amino acids at the CDT1-geminin interface, with a focus on the CDT1 region 3.2 loop (Figure 5i). Mutation of CDT1 residues 327-330 to alanine (CDT1^{4A}) resulted in a CDT1 construct that can load MCM2-7, but was resistant to geminin inhibition (Figure 5l-m), and displayed an interaction with geminin (Figure 5j). R330 had the highest $\Delta\Delta G$ score in the BUDE alanine scan, and its mutation to alanine had the same impact as CDT1^{4A}, highlighting its importance for the CDT1 geminin interaction (Figure 5k-m).

5. CDK/Cyclin-Mediated Inhibition of Licensing CDK/cyclin complexes phosphorylate multiple components of the preRC, not just Cdt1. Therefore, the inhibition observed in Figure 8 may result from CDK-mediated phosphorylation of several preRC factors, not solely Cdt1. The authors should specifically address whether CDK-dependent phosphorylation of Cdt1 alone is sufficient to inhibit licensing in the presence of geminin. Clarifying the contribution of Cdt1 phosphorylation relative to other CDK targets would improve the mechanistic interpretation of the presented model.

This was an excellent suggestion. Thus, we have performed an assay in which we have pre-phosphorylated individual pre-RC proteins to determine their effect on DH loading in the pre-RC assay (Figure 8c and d). This assay highlighted that ORC1-5 and CDC6 primarily reduce MCM2-7 loading when phosphorylated. We found that the combination of ORC1-5 and CDC6 phosphorylation and sequestration of CDT1 by geminin work in synergy to inhibit helicase loading (Figure 8e and f). We have updated the text to reflect these findings.

6. Confusing Nomenclature for Cdt1 Constructs

The designation of the truncated version simply as “Cdt1” is both careless and misleading. According to standard conventions, the unmodified (wild-type) protein should be labeled as “Cdt1” or “FL” when used in conjunction with truncated constructs, while truncations should be named based on the specific deleted domain. It is important to clarify this nomenclature to prevent confusion among readers. For instance, the manuscript refers to the "N-terminus" of the

truncated “Cdt1” construct, although this region actually corresponds to the middle helical domain (MHD) of the wild-type or full-length Cdt1 protein.

We appreciate the comment, which helps provide greater clarity. We have re-labelled all figures and changed the text using the terms CDT1^{ΔN} and CDT1^{FL}. We only use “CDT1” when making generic observations about the protein. We have also ensured that the correct reference is made to the MHD versus the N-terminus.

Minor Points

- In the presentation of the preRC assay results, the names of the individual proteins should be clearly indicated.

We have now added labels to all gels, except for Fig. 1j where there is insufficient space.

- Remove duplicated conclusions. One paragraph states: "As such, we define the minimal geminin coiled-coil section that is required for CDT1 binding as residues 96–135." The following paragraph repeats: "Subsequently, we define Gem96–135 as the minimal region of geminin that is capable of binding to CDT1." These should be consolidated or rephrased to avoid redundancy.

We have deleted the repeated conclusion.

- Use the term coiled-coil rather than coil-coil throughout the manuscript.

This has been corrected.

- In Figure 1j, the input lanes should also show the 5× loading condition for comparison.

We have now added these extra loads to the figure.

- In Figure 7a, use distinct colours to clearly highlight the geminin residues that sterically clash with Mcm2.

We have made this requested change.

Reviewer #3

This study investigates the inhibitory role of Geminin during the replication licensing process. Utilizing their established system with purified human proteins, the authors address this important question through a biochemical approach. They examine how Geminin affects various steps in replication licensing, including the recruitment and loading of MCM2-7, and seek to characterize the oligomeric state and structural domains of the Cdt1-Geminin complex required for inhibition. While the central claim of the paper is supported by the data, there are notable conceptual and experimental limitations that constrain the study's potential impact and contribution to the field.

Conceptually, certain results from the licensing assay are presented in a way that may be confusing or unclear, potentially making it difficult for readers to fully follow the narrative. Technically, while some of the most intriguing and novel findings are under explored, the manuscript places disproportionate emphasis on negative or less relevant results. Together, these issues somewhat obscure the clarity and diminish the perceived novelty of the study's overall message.

We thank the reviewer for their thoughtful response and have worked to address the points they raised.

The claim that “the mechanism of this inhibition has remained unknown for the last 25 years” is perhaps a bit too strong.

As suggested, we have revised the wording.

1. There is insufficient rationale or interpretation of the experiments in Figure 1 i-l that mix the individual domains of CDT1 – what do the authors think is happening in those reactions, and is the effect meaningful? Does it suggest anything about what human CDT1 does in the human preRC reactions?

Many thanks for the suggestion. We have added a conclusion in respect of CDT1's function in DNA licensing and the geminin inhibition mechanism. We highlight that CDT1 double-tethering to MCM2-7, involving two weak interactions, allows for easy release of CDT1, as breaking one of the two connections is sufficient to disrupt the CDT1-MCM2-7 complex. This seems important, as the permanent binding of CDT1 via a single strong CDT1-MCM2-7 interaction could potentially block helicase activation factors from binding to MCM2-7. Also, this mechanism allows for easy

inhibition by geminin for two reasons: 1.) only one of two CDT1 interactions with MCM2-7 needs to be blocked to impair pre-RC formation. 2.) Splitting the CDT1-MCM2-7 interface into two interactions allows for relatively weak interactions between CDT1 and MCM2-7, which can be more easily broken by geminin.

2. It isn't clear what the importance of the CDT1 "region 1" helix predicted by AlphaFold3 is since there is no detectable difference between full-length CDT1 and a version lacking the entire N-terminus. The authors devote 2 main figures to this predicted interaction, but do not show that it occurs or is consequential (in vitro). Alternatively, the authors should state if they think this predicted interaction is real, and if so, what it might be for.

We thank the reviewer for the comment. Our study benefits from a systematic approach, providing a complete analysis that will serve as a reference for the field. That said, we understand the reviewer's concern and have therefore consolidated figures 3 and 4 to devote less space to the AlphaFold analysis. We have also added to the concluding sentence on Region 1: "the predicted positioning of this helix by AlphaFold is likely charged-based (Supplementary Figure 8b-e)" .

3. Another separation of CDT1 binding from geminin function allele that changes two amino acids was described by Suchyta et al (PMID: 25988259). How does this allele fit with the authors' model?

Many thanks for the excellent question. These mutations sit at the interface of both coiled coils and at the hinge regions, thus affecting two different CDT1-geminin interactions. AlphaFold 3 predictions of this mutant highlight a structural re-organisation of region 3.2 (Supplementary Figure 21), potentially disrupting several CDT1-geminin interactions in this region. We suggest that these changes have the potential to alter the relative location of the coil-coil, potentially removing the geminin-Mcm2 clash. We have added this information to the discussion.

4. One experiment to try to directly test the clash model predicted by AlphaFold3 is in supplemental Figure 17. Although creative, I'm not convinced this glycine insertion demonstrates that the angle of geminin relative to MCM is the critical molecular feature. The argument is that CDT1 "positions the geminin coiled-coil at the correct angle to generate the geminin-MCM2 clash." The LINK mutant binds CDT1 about 3 times less well compared to wild-type, and there's no experiments that test the angle or that CDT1 itself uses the predicted binding site to create that angle. The model remains speculative and should be presented as such.

We appreciate the comment. We revised the language and state: “data strongly suggest that both of geminin’s CDT1 binding sites act in coordination and hint that this coordination could be necessary for the positioning of the coiled-coil of geminin.” We also note that our analysis of the AWA mutant described by Suchyta et al (PMID: 25988259) identified that the mutations are located at the interface of coiled-coil and region 3.2 (Supplementary Figure 21), which also fits with the concept we presented.

5. The dissection of the geminin coiled-coil region is only through changing the length and not the sequence. There is a strong, yet unacknowledged, assumption that the specific amino acids are irrelevant and only the length matters. If the interpretation is correct, then any sequence that forms a coiled-coil could substitute. The authors should address this point in some way.

Geminin has a non-canonical coiled-coil, as the two coils do not have variable angles and twists. As such, it is not straightforward to replace it. We have inserted a short linker, which is predicted to maintain the coiled-coil but alters the relative location of the coiled-coil. This mutant was still fully functional in inhibition (Supplementary Figures 19e-h). Moreover, we analysed the coiled-coil sequence conservation. The clash region is relatively poorly conserved at the amino acid identity level, while amino acid similarity is higher (Supplementary Fig. 17c). Key amino acids that stabilise the coiled coil are conserved, while other parts are less conserved (Supplementary Fig. 17c and d). Based on this analysis, the specific amino-acid sequence of the coiled-coil appears less critical for its function. We have update the manuscript accordingly.

Minor:

a) Is the length of the geminin coiled-coil region that is predicted to clash with MCM2 conserved?

We have performed a sequence alignment, and it shows that the overall length of the coiled coil is conserved (Supplementary Fig. 17c).

b) A brief discussion of the limitations of the study would be helpful for future readers. For example CDT1 and geminin were produced in bacteria, but both proteins are reported to be decorated with post-translational modifications in cells (PhosphoSitePlus), the loading reactions use purified DNA, etc.

This is another excellent point, and we added this to the discussion.

Response to Reviewers comments:

Reviewer #1 (Remarks to the Author):

The revised manuscript from Speck and colleagues has done an excellent job of responding to my concerns. The addition of the experiments examining the effects of CDK phosphorylation of individual origin licensing proteins greatly improves the last part of the paper. The changes in the text in response to the reviews has also greatly clarified the manuscript. This work will be an excellent addition to the literature and of interest to those studying the DNA replication, the cell cycle and the molecular mechanisms of complex assembly events in the cell.

Author response:

We sincerely thank the reviewer for the very positive and encouraging comments. We are pleased that the additional experiments and revisions have strengthened and clarified the manuscript.

Reviewer #2 (Remarks to the Author):

Despite addressing most of the points previously raised by this reviewer in a positive manner, I still have significant concerns regarding the superficial treatment of some of the novel findings, as well as the disproportionate emphasis placed on results that are less relevant to the main conclusions of the study, as also noted by Reviewer 3.

Regarding the lack of controls for novel findings. The question of how CDK regulates licensing in human cells remains one of the most critical and incompletely understood issues in the field, as it is essential for proper regulation of chromosome replication. Therefore, if the authors choose to address this topic, it must be done with appropriate rigor, including, at a minimum, phosphorylation controls. In Figure 8, the authors present results from pre-incubating pre-RC components with CDK; however, they provide no evidence confirming that these proteins were actually phosphorylated. Moreover, they do not assess the effectiveness of the CDK inhibitor, which is essential to ensure that the CDK added later in the reaction is fully inhibited. These missing controls are critical for meaningful interpretation of the results and to avoid further misunderstanding in the field.

Author response:

We appreciate the reviewer's thoughtful and constructive feedback. We agree that additional controls would strengthen the study and have now included these data in Supplementary Figure 21. Specifically, we performed western blot analyses using a phospho-SP antibody and detected cyclin A-CDK1-dependent phosphorylation of ORC1, CDC6, and CDT1, while ORC6 was not phosphorylated. MCM2-7 showed basal phosphorylation that increased upon CDK1 treatment. Importantly, in the presence of p27, no phosphorylation was detected, confirming complete CDK inhibition. CDC6 exhibited the strongest phosphorylation signal, approximately 100-fold greater than any other protein. Using cyclin A-CDK2, we observed similar results, although CDT1 and MCM2-7 were not phosphorylated under these conditions. These additional data confirm both the phosphorylation of target proteins and the effectiveness of CDK inhibition, thereby addressing the reviewer's concern. The manuscript was checked to reflect these changes.

Continued - Reviewer #2 (Remarks to the Author):

Concerning the disproportionate emphasis on less relevant results. The geminin deletions and/or Cdt1 point mutants described in Figure 5, the only structural or AlphaFold-based figure showing positive results, could have been designed in exactly the same way using the crystal structure published by Lee et al., Nature 2004, which clearly identifies all key residues involved in the interaction. Therefore, the sentence in the abstract—“AlphaFold modelling revealed a critical N-terminal CDT1-binding helix and loop in geminin, which proved essential for inhibition”—while appealing, should be removed. If the authors choose to retain it, they must clearly state that these findings are consistent with the previously published structural data by Lee et al., to avoid overstating the novelty of their results.

Author response:

We appreciate the reviewer’s detailed assessment and the opportunity to clarify this point. Accordingly, we modified the sentence in the abstract to the following wording “AlphaFold modelling **provided structural insights into an** N-terminal CDT1-binding helix **of** geminin, which proved essential for inhibition”.

We did not delete the sentence for the following reasons: The study by Lee et al. (2004) used mouse proteins, and our analysis revealed that a functionally critical region of human geminin has a different organisation from the corresponding segment in the mouse protein. The divergence likely results from structural differences near the C-terminus of the crystallised construct or from crystal packing effects.

Furthermore, our data (Wells et al Nature Communications 2025) clearly demonstrate that hCDT1 and hMCM2-7 do not form a complex in the absence of hORC-hCDC6-DNA, whereas Lee et al. reported an interaction between mCDT1 and mMCM4/6/7 without ORC, CDC6, or DNA, which geminin was said to inhibit. We cannot reconcile these findings, suggesting either species-specific differences or experimental discrepancies.

Moreover, Lee et al. described a geminin-regulated interaction between mMCM4/6/7 and mCDT1, which seems inconsistent with more recent data (our study and Wells et al., Nature Communications 2025) showing that CDT1 interacts with MCM2 in the context of the OCCM, and that geminin inhibits this MCM2-CDT1 interaction. The mMCM4/6/7 complex used in Lee et al. lacks MCM2, making their proposed mechanism difficult to reconcile with our biochemical and structural observations.

Taken together, we believe that our AlphaFold-based analysis of human geminin and CDT1 provides valuable new insights specific to the human system and extends, rather than duplicates, earlier work.

Reviewer #3 (Remarks to the Author):

The revised version is improved and satisfies our requests. The authors should be congratulated.

Author response:

We thank the reviewer for the positive feedback and are pleased that the revised manuscript meets expectations.